# Enteric pathogens deploy cell cycle inhibiting factors to block the bactericidal activity of Perforin-2

Ryan M McCormack, Kirill Lyapichev, Melissa L Olsson, Eckhard R Podack*[†], George P Munson*[†]

Department of Microbiology and Immunology, Miller School of Medicine, University of Miami, Miami, United States

**Abstract** Perforin-2 (MPEG1) is an effector of the innate immune system that limits the proliferation and spread of medically relevant Gram-negative, -positive, and acid fast bacteria. We show here that a cullin-RING E3 ubiquitin ligase (CRL) complex containing cullin-1 and βTrCP monoubiquitylates Perforin-2 in response to pathogen associated molecular patterns such as LPS. Ubiquitylation triggers a rapid redistribution of Perforin-2 and is essential for its bactericidal activity. Enteric pathogens such as *Yersinia pseudotuberculosis* and enteropathogenic *Escherichia coli* disarm host cells by injecting cell cycle inhibiting factors (Cifs) into mammalian cells to deamidate the ubiquitin-like protein NEDD8. Because CRL activity is dependent upon NEDD8, Cif blocks ubiquitin dependent trafficking of Perforin-2 and thus, its bactericidal activity. Collectively, these studies further underscore the biological significance of Perforin-2 and elucidate critical molecular events that culminate in Perforin-2-dependent killing of both intracellular and extracellular, cell-adherent bacteria.

*For correspondence: epodack@ miami.edu (ERP); gmunson@ miami.edu (GPM)

[†]These authors contributed equally to this work

## Introduction

As the largest class of ubiquitin ligases, cullin-RING E3 ubiquitin ligases (CRLs) regulate numerous cellular processes including signal transduction, gene expression, development, and cell cycling (*Bosu and Kipreos, 2008*; *Metzger et al., 2012*). CRLs are modular complexes that are assembled from a profusion of subunits. However, most share a similar architecture. At the core of each lies an elongated cullin upon which other CRL subunits assemble (*Bosu and Kipreos, 2008*). Adaptor molecules bind to the cullin's extended amino-terminal domain and are themselves bound by receptors that provide substrate specificity (*Wu et al., 2003*; *Cardozo and Pagano, 2004*; *Petroski and Deshaies, 2005*; *Lydeard et al., 2013*). The RING subunit binds to the cullin's globular carboxy-terminal domain and acts as an E3 ubiquitin ligase responsible for recruiting the complex's ubiquitin conjugating enzyme (E2). The placement of the substrate and E2 at opposite ends of the elongated cullin translates into a separation of ~50 Å (*Wu et al., 2003*; *Hao et al., 2007*; *Merlet et al., 2009*). This gap prohibits ubiquitylation of the substrate. This problem is solved by an additional E2 enzyme, such as UBC12, that conjugates NEDD8, an 8.6 kDa member of the ubiquitin family of proteins (UniProt entry Q15843, Pfam identifier PF00240), to a conserved lysine within the carboxy-terminal domain of the cullin (*Petroski and Deshaies, 2005*). Cullin neddylation induces a conformational change that places the ubiquitin E2 and substrate in sufficient proximity for ubiquitylation to occur (*Duda et al., 2008*; *Saha and Deshaies, 2008*). Thus, CRL-dependent ubiquitylation of a protein substrate is itself dependent upon cullin neddylation (*Morimoto et al., 2000*; *Ohh et al., 2002*; *Sakata et al., 2007*).

Cycle inhibiting factors (Cifs) are bacterial effector proteins that inactivate CRLs through deamidation of NEDD8 (*Cui et al., 2010*; *Boh et al., 2011*; *Crow et al., 2012*). They are delivered

**eLife digest** A wide range of bacteria and other microbes can infect animals and cause disease. Throughout evolution, these microbes and their hosts have been fighting never ending arms races in which the microbes deploy ever more elaborate weapons, while the hosts adapt to defend themselves. An animal's first line of defense is provided by its 'innate' immune system. This system is activated by the general features of microbial cells; for example, the molecules that make up the walls surrounding most bacteria. Microbes must defeat the innate immune system in order to cause disease, and ultimately to spread from one host to the next.

One component of innate immunity is a protein called Perforin-2 that is present in most, if not all, animal cells. This protein forms pores on bacterial cells, causing them to split open and die. However, it was not clear how Perforin-2 is switched on and what, if anything, bacteria do to counteract it. To address these questions, McCormack et al. infected human and mice cells with bacteria that cause serious diseases of the digestive tract.

The experiments show that when animal cells detect bacteria, or merely a fragment of their cell wall, a specific group of proteins, called the CRL complex, attaches a molecule called ubiquitin to Perforin-2. Ubiquitin works much like the shipping label of a package, enabling the efficient targeting of Perforin-2 to the invading bacteria. McCormack et al. also show that some bacteria use a protein called a cell cycle inhibiting factor (or Cif for short) to inhibit the CRL complex. This blocks the ubiquitin labeling of Perforin-2, which renders it a useless weapon that can no longer be directed towards bacteria.

Mice that are infected with a bacterium called *Yersinia pseudotuberculosis* become seriously unwell and often die. However, McCormack et al. found that mice infected with mutant *Y. pseudotuberculosis* that lacked Cif remained healthy. Also, mice that lacked Perforin-2 are highly susceptible to infectious diseases. McCormack et al.'s findings reveal how Perforin-2 is activated during the innate immune response and how some bacteria can defeat this pivotal defense. In the current age of antibiotic resistant bacteria, these studies may spur the development of new drugs that restore or increase the activity of Perforin-2.

to the cytosol of eukaryotic cells by type III secretion systems of some Gram-negative pathogens including *Yersinia pseudotuberculosis* and enteropathogenic *Escherichia coli* (EPEC) (*Marches et al., 2003*; *Charpentier and Oswald, 2004*; *Jubelin et al., 2009*; *Taieb et al., 2011*). Upon entering the cytosol Cifs proceed to deamidate Gln40 of NEDD8 thus producing a Glu residue at that position (*Cui et al., 2010*). Because CRL activity is dependent upon NEDD8, this enzymatic modification prevents the ubiquitylation of CRL substrates (*Marches et al., 2003*; *Saha and Deshaies, 2008*; *Toro et al., 2013*). The discovery of Cif and elucidation of its enzymatic mechanism can be traced back to initial reports that certain pathogens cause cell cycle arrest (*De Rycke et al., 1997*; *Nougayrede et al., 2001*). It is now known that Cif causes the accumulation of cell cycle inhibitors by blocking their ubiquitylation and subsequent degradation by the 26S proteasome (*Marches et al., 2003*; *Taieb et al., 2006*; *Samba-Louaka et al., 2008*). It has been proposed that Cif mediated cell cycle arrest provides enteric pathogens, such as EPEC and *Y. pseudotuberculosis*, with a stable platform by decreasing the turnover rate of intestinal epithelial cells (*Samba-Louaka et al., 2009*). In theory this mechanism could promote colonization of the gastrointestinal tract. Although this is certainly a well reasoned hypothesis, to the best of our knowledge it remains untested. In addition, CRLs regulate diverse cellular processes. Although the majority of these processes probably have no role in limiting pathogen virulence, others may. Therefore, it is not yet known how the deamidation of NEDD8 by Cif contributes to pathogenicity.

Recent studies suggest that Perforin-2 (macrophage-expressed gene 1; MPEG1) is an effector of the innate immune system that limits the proliferation and spread of medically relevant Gram-negative, -positive, and acid fast bacteria. For example, expression of Perforin-2 in murine embryonic fibroblasts (MEFs) is associated with the killing of intracellular *Salmonella enterica* serovar Typhimurium (hereafter *Salmonella typhimurium*), methicillin-resistant *Staphylococcus aureus* (MRSA), *Mycobacterium smegmatis, and Mycobacterium avium* (*McCormack et al., 2013b*). Moreover, siRNA knockdown of Perforin-2 expression abolished the ability of MEFs to destroy intracellular bacteria

unless the siRNA transfected cells were also complemented with a siRNA resistant Perforin-2-RFP expression plasmid (*McCormack et al., 2013b*). More recently, knockdown of Perforin-2 allowed *Chlamydia trachomatis*, an obligate intracellular pathogen that normally replicates within epithelial cells, to proliferate within mammalian macrophages (*Fields et al., 2013*). In vivo, Perforin-2 is critical for protection against MRSA and *S. typhimurium* (*McCormack et al., 2015*).

Additional studies by McCormack et al. suggest that Perforin-2 has a primary role in the destruction of bacterial pathogens (*McCormack et al., 2015*). Nevertheless, Perforin-2 remains a poorly characterized molecule. To address this deficiency, we sought to characterize the molecular events required for the activation and deployment of Perforin-2 through a variety of in vitro and in vivo infection models. In this study we show that Perforin-2 is ubiquitylated by a CRL complex containing CUL1 and βTrCP in response to infectious bacteria or pathogen associated molecular patterns (PAMPs) such as LPS. Ubiquitylation triggers a rapid reorganization of Perforin-2-RFP within the cytosol and is absolutely required for its bactericidal activity. Due to its ability to inactivate CRLs, the bacterial effector protein Cif blocks ubiquitylation of Perforin-2 and its subsequent redistribution within infected cells. This blockade prevents Perforin-2-dependent killing of bacteria in vitro. In vivo, wild-type *Y. pseudotuberculosis* is significantly more virulent than a *cif* mutant. This difference was not observed with Perforin-2 deficient mice which succumb to an otherwise non-lethal dose of either wild-type or Cif⁻ bacteria. With regards to pathogenicity, these latter results suggest that the inhibition of Perforin-2-dependent killing is the primary function of Cif.

## Results

### Perforin-2 is ubiquitylated and co-immunoprecipitates with components of a CRL complex

Perforin-2 contains an amino-terminal membrane attack complex perforin (MACPF) domain (*Figure 1*). This domain is also present in the pore-forming components of complement and Perforin-1 (*Podack and Tschopp, 1982*; *Dennert and Podack, 1983*; *Podack and Dennert, 1983*; *DiScipio et al., 1984*; *Lichtenheld et al., 1988*). Moreover, crystallographic studies of MACPF domains have revealed structural similarities to bacterial pore-forming CDCs (cholesterol dependent cytolysins) (*Rosado et al., 2007*; *Rosado et al., 2008*; *Slade et al., 2008*; *Law et al., 2010*). Thus, the presence of a MACPF domain within Perforin-2 suggests that it may also have the ability to form lytic pores. This hypothesis is supported by the recovery of Perforin-2 from bacteria and images of pores with an average diameter of 100 Å in lipid membranes of mammalian cells expressing Perforin-2 and bacterial cell walls exposed to Perforin-2 (*McCormack et al., 2015*). The MACPF domain of Perforin-2 is immediately followed by a domain of unknown function (*Figure 1*). We have dubbed this domain the Perforin-2 (P2) domain because it has been conserved throughout evolution in Perforin-2 orthologs and is only associated with the MACPF domain of Perforin-2. Although the function of the Perforin-2 domain remains to be elucidated, its conservation suggests that it is essential. The presence of a putative transmembrane alpha helix is an additional feature that distinguishes Perforin-2 from Perforin-1 and the pore-forming components of complement (*Figure 1*). Topological modeling and sequence analyses indicate that Perforin-2 is a type I transmembrane protein with a relatively short carboxy–terminal domain (*McCormack et al., 2013b*). The localization of Perforin-2 to vesicles staining with markers for endoplasmic reticulum, Golgi, early endosomes, and plasma membrane is consistent with this prediction (*McCormack et al., 2015*).

The topology of a type I transmembrane protein would orient the MACPF domain of Perforin-2 towards bacteria contained within the lumen of membrane vesicles or towards extracellular bacteria that adhere to the plasma membrane. This topology would also place Perforin-2's 40 amino acid long carboxy-terminal domain in the cytosol. Therefore, we hypothesized that regulatory proteins may interact with the carboxy-terminal domain of Perforin-2 to govern its bactericidal activity. To identify these proteins we used Perforin-2's carboxy-terminal domain as bait for macrophage-expressed gene products in a yeast two-hybrid system and identified UBC12 as a Perforin-2-interacting protein. We then confirmed this interaction by coimmunoprecipitation of UBC12 with Perforin-2-GFP (*Figure 2A*). The addition of GFP, via a flexible linker, to the carboxy terminus of Perforin-2 was necessary because antibodies that immunoprecipitate native Perforin-2 are not available (*McCormack et al., 2013a*, *2013b*). However, this also allowed us to use GFP as a negative specificity control and, as expected, UBC12 did not coimmunoprecipitate with GFP (*Figure 2A*).

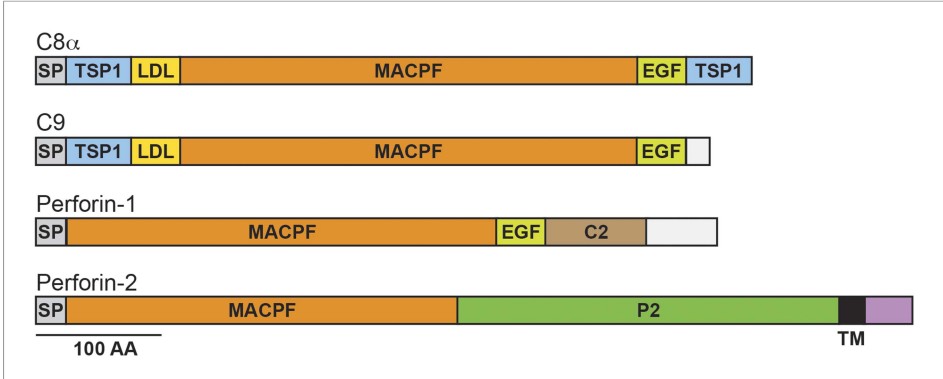

**Figure 1**. Domain organization of human C8a, C9, Perforin-1, and Perforin-2. The pore forming components of complement (C8α and C9), Perforin-1, and Perforin-2 all contain membrane attack complex perforin (MACPF) domains and amino-terminal signal peptides (SPs). MACPF domains are also present in other components of complement (C6, C7, and C8β). The presence of a MACPF domain within Perforin-2 suggest that it is also a mediator of innate immunity. Unlike C6-C9 and Perforin-1, Perforin-2 is predicted to be an integral membrane protein because it alone contains a membrane spanning alpha helix (TM) followed by a short cytosolic tail. An additional distinguishing feature of Perforin-2 is the P2 domain which is of unknown function but conserved amongst Perforin-2 orthologs. Domain architecture was retrieved from UniProt entries P07357, P02748, P14222, and Q2M385. TSP1, thrombospondin type-1 repeat; LDL, low-density lipoprotein receptor class A repeat; EGF, epidermal growth factor-like domain; C2, calcium-dependent phospholipid binding domain.

UBC12, also known as UBE2M, is an E2 enzyme that conjugates NEDD8 to cullins of multi-component CRLs (*Huang et al., 2005*, *2009*). Therefore, we also probed coimmunoprecipitates for other CRL components that associate with Perforin-2 and found that CUL1 (cullin-1) and βTrCP coimmunoprecipitate with Perforin-2-GFP but not with GFP (*Figure 2A*). As with other cullins, CUL1 is a CRL scaffold. F-box proteins such as βTrCP1 and -2 provide substrate specificity (*Huang et al., 2005*, *2009*). Because these molecular associations suggest that Perforin-2 is ubiquitylated by a CRL complex, we next sought to determine whether or not Perforin-2 is ubiquitylated. As reported

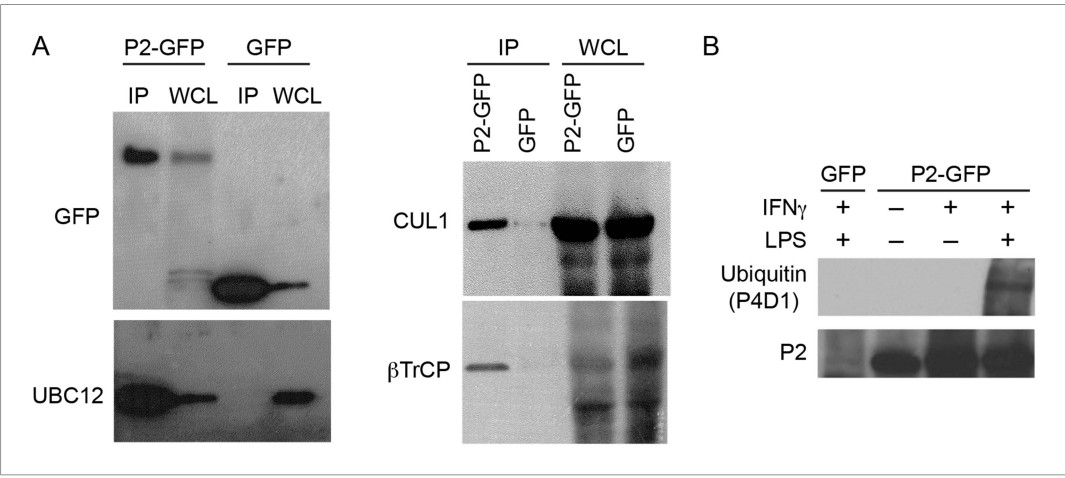

**Figure 2**. Perforin-2 co-immunoprecipitates with CRL subunits and is ubiquitylated. RAW 264.7 cells were transfected with Perforin-2-GFP or GFP expression constructs and stimulated with IFN-γ prior to immunoprecipitation with anti-GFP. (**A**) Western blots probed with the indicated antibodies reveal that the NEDD8 E2 ligase UBC12, CRL cullin scaffold CUL1, and substrate receptor F-box protein βTrCP specifically coimmunoprecipitate with Perforin-2. (**B**) Additional western blots reveal that IFN-γ and LPS, but not IFN-γ alone, stimulate ubiquitylation of Perforin-2. P2, Perforin-2; IP, immunoprecipitates; WCL, whole cell lysates.

previously, IFN-γ induces the expression of Perforin-2 (*Fields et al., 2013*; *McCormack et al., 2013b*). However, ubiquitylation of Perforin-2 does not occur unless a PAMP, such as LPS, is also present (*Figure 2B*). The ubiquitylation of Perforin-2 in response to a PAMP suggest that ubiquitylation may be essential for its bactericidal activity.

## Perforin-2 bactericidal activity is dependent upon ubiquitylation

To examine the biological relevance of CRL subunits that coimmunoprecipitate with Perforin-2 we used siRNA to ablate the expression of UBC12 and CUL1 (*Figure 3A*). As expected from our coimmunoprecipitations, siRNA knockdown of either UBC12 or CUL1 blocked ubiquitylation of Perforin-2 (*Figure 3B*). Although βTrCP also coimmunoprecipitated with Perforin-2, there are two mammalian βTrCP paralogs (βTrCP1 and -2) with overlapping activities. Therefore we did not target βTrCP due to the difficulty of simultaneously silencing both genes.

We next transfected CMT93 cells, a murine intestinal epithelial cell line, with a variety of siRNAs and expression plasmids to evaluate their effects in a bactericidal assay. Since the pathogen chosen for these assays, EPEC, is primarily extracellular we also chose to investigate extracellular killing rather than the intracellular killing that has been reported previously (*Fields et al., 2013*; *McCormack et al., 2013b*). Killing of extracellular bacteria was evident within 2–4 hr after attachment to cells transfected with control scramble siRNA (*Figure 3C–E*). Killing was reduced or eliminated by siRNA knockdown of Perforin-2, CUL1, or UBC12. Moreover, siRNA knockdown of either CUL1 or UBC12 blocked Perforin-2-dependent killing of EPEC as efficiently as Perforin-2 siRNA (*Figure 3D,E*). Killing was restored

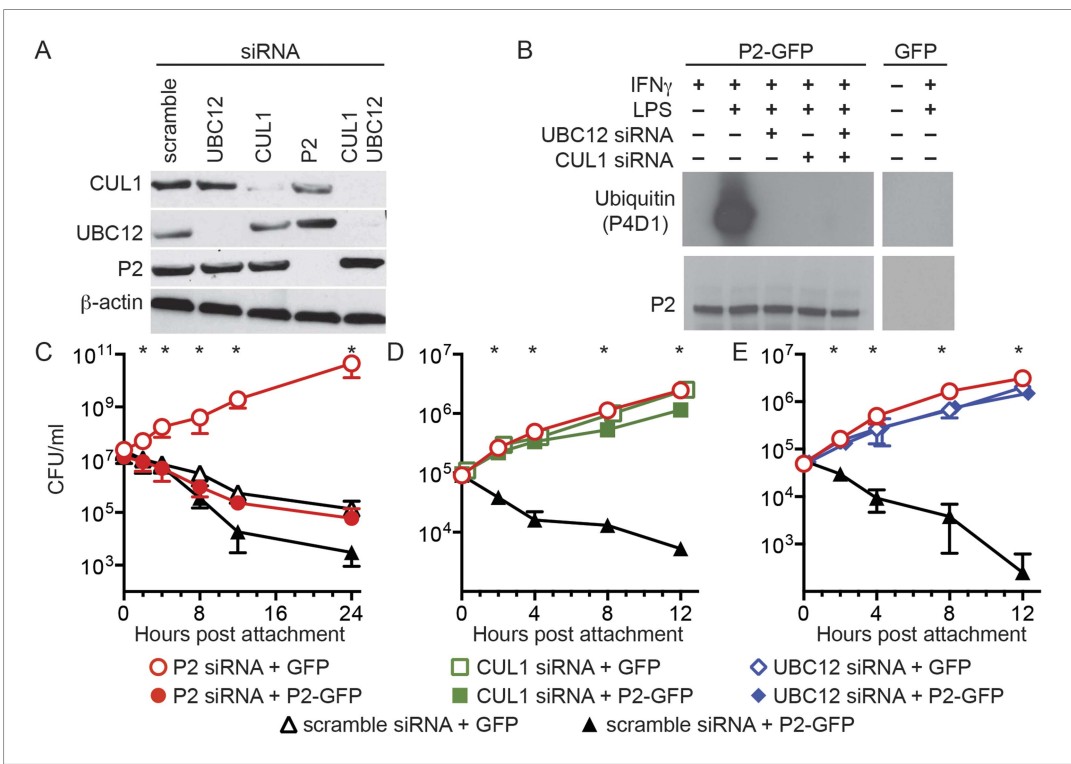

**Figure 3**. Perforin-2 activity is dependent upon its ubiquitylation by a CRL complex. (**A**) siRNA efficiently and specifically knocks down expression of CUL1, UBC12, and Perforin-2 in IFN-γ stimulated CMT93 cells. (**B**) Western blots of immunoprecipitates from transfected CMT93 cells probed with the indicated antibodies demonstrate that Perforin-2 is ubiquitylated in cells stimulated with IFN-γ and LPS and that ubiquitylation is dependent upon UBC12 and CUL1. (**C–E**) In contrast to scramble siRNA, siRNA knockdown of Perforin-2, CUL1, or UBC12 in CMT93 cells promotes survival of extracellular E2348/69, an enteropathogenic *Escherichia coli* (EPEC) strain that does not express cycle inhibiting factor (Cif). Solid and open symbols denote cotransfection with Perforin-2-GFP or GFP expression plasmids, respectively. *Statistically significant (p < 0.05) differences between (**C**) Perforin-2 siRNA + GFP and the other three conditions, (**D**, **E**) scramble siRNA + Perforin-2-GFP and the other three conditions by one-way ANOVA with Bonferroni multiple-comparisons post-hoc test; *n* ≥ 3. P2, Perforin-2.

in cells transfected with Perforin-2 siRNA by cotransfection of a siRNA-insensitive Perforin-2-GFP expression plasmid (*Figure 3C*). However, Perforin-2-GFP could not overcome the killing defect of CUL1 or UBC12 ablated cells (*Figure 3D,E*). These results demonstrate Perforin-2 cannot kill bacteria in the absence of ubiquitylation.

The carboxy terminal tail of Perforin-2 is the most likely site of ubiquitylation based on topological modeling (*Figure 1*). Human and mouse Perforin-2 both contain a cluster of four lysine residues; three of which are highly conserved across mammalian species (*Figure 4A*). As predicted, site directed mutagenesis of the three most highly conserved lysines abolished ubiquitylation of Perforin-2-GFP (*Figure 4B*). However, the triple mutation did not affect protein expression nor stability (*Figure 4B*). *Mpeg1* knockout MEFs transfected with a plasmid that expresses the triple mutant or GFP were

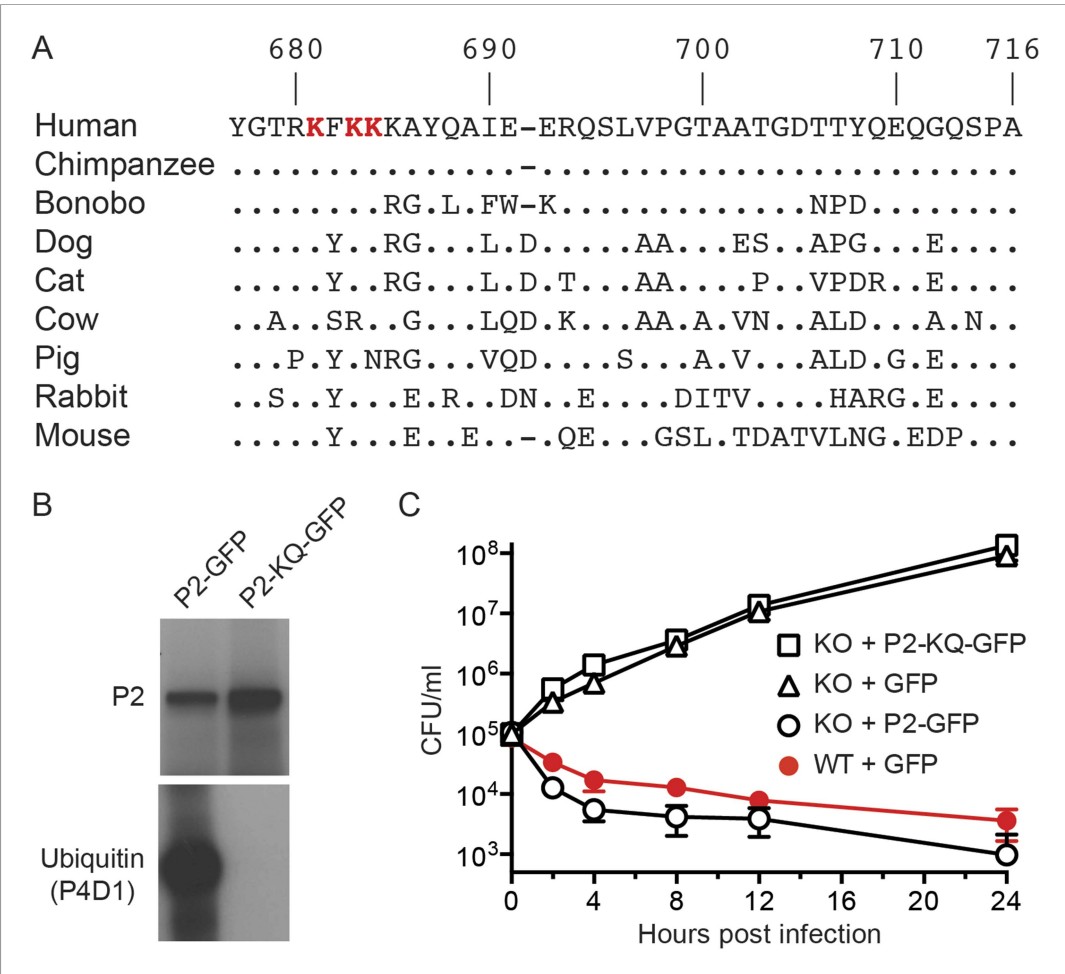

**Figure 4**. Conserved lysines within the carboxy-terminal domain of Perforin-2 are required for its ubiquitylation and bactericidal activity. (**A**) Alignment of the predicted cytosolic domains of Perforin-2 (P2) molecules from select mammalian species. Dots denote identity to human Perforin-2. The three most conserved lysine residues are highlighted in red. Numbering is relative to human Perforin-2, GenBank accession number AAI12231. (**B**) Site-directed mutagenesis was used to mutate the three conserved lysines to glutamine residues in murine Perforin-2. Western blots of Perforin-2-GFP or Perforin-2-KQ-GFP expressed in CMT93 cells stimulated with IFN-γ and LPS demonstrate that both fusion proteins are expressed. However, only Perforin-2-GFP is ubiquitylated. (**C**) Murine embryonic fibroblasts (MEFs) isolated from wild-type and Perforin-2 −/− (KO) embryos were transfected with GFP, Perforin-2-GFP, or Perforin-2-KQ-GFP expression plasmids. Transfected cells were induced with IFN-γ ca. 24 hr before infection with *Salmonella typhimurium*. The results demonstrate that Perforin-2-GFP, but not the K-to-Q mutant nor GFP, restored killing in KO MEFs to wild-type levels. The differences between KO MEFs transfected with Perforin-2-GFP and Perforin-2-KQ-GFP or GFP are statistically significant, $p < 0.05$, at hours 2 through 24 as determined by one-way ANOVA with Bonferroni post-hoc multiple comparisons; $n \geq 3$.

unable to kill intracellular *S. typhimurium* (**Figure 4C**). In contrast, knockout MEFs transfected with a Perforin-2-GFP expression plasmid cleared the pathogen as efficiently as wild-type MEFs. In aggregate, these results demonstrate that ubiquitylation of Perforin-2 is essential for Perforin-2-dependent killing of both extra- and intracellular bacteria. Moreover, the ubiquitylation of one or more conserved lysines within the carboxy-terminal tail of Perforin-2 is accomplished by a CRL complex in response to bacterial antigens.

## Enteric pathogens block ubiquitylation of Perforin-2

Since we have shown above that Perforin-2 is a CRL substrate, we evaluated the ability of Cif[+] and Cif[−] pathogens to block ubiquitylation of Perforin-2. As expected, wild-type *Y. pseudotuberculosis* blocked ubiquitylation of Perforin-2 whereas an isogenic *cif::aadA* mutant did not (**Figure 5**). Transformation of the *cif* mutant with a Cif expression plasmid restored the pathogen's ability to block

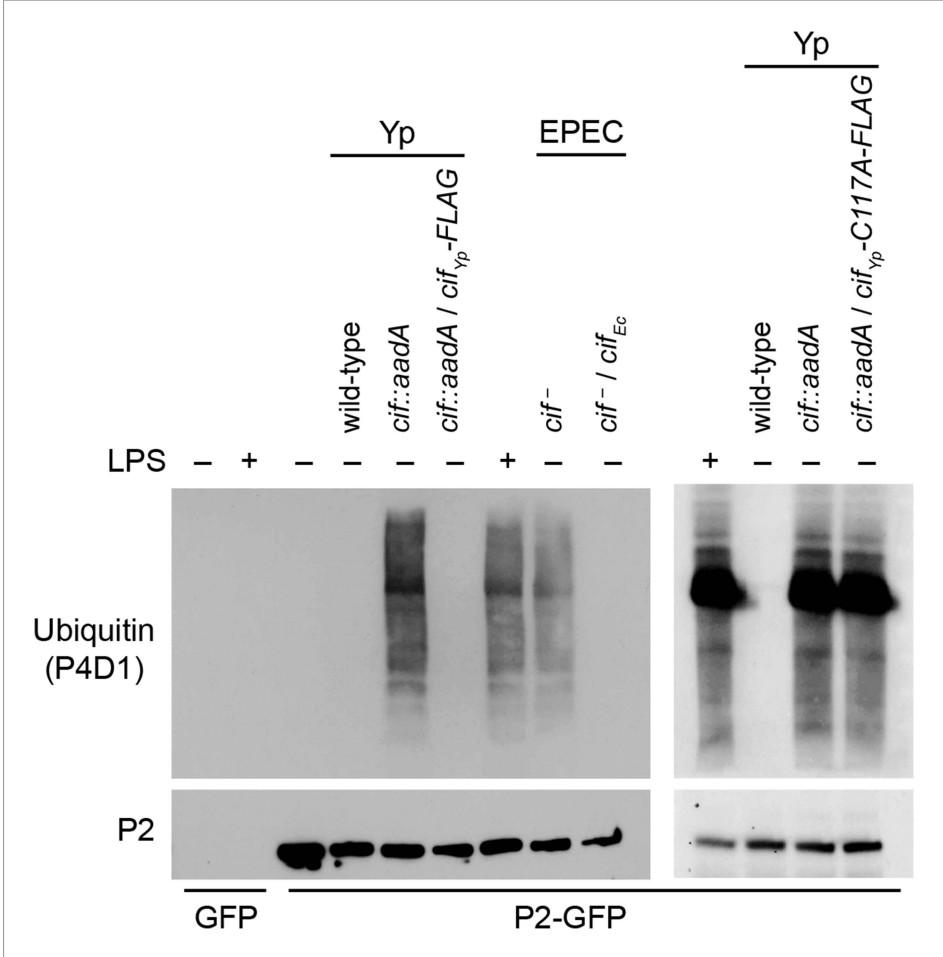

**Figure 5**. Cif blocks ubiquitylation of Perforin-2. CMT93 cells were transfected with GFP or Perforin-2-GFP expression plasmids and stimulated with IFN-γ. As indicated, some transfected cells were also stimulated with LPS, infected with *Yersinia pseudotuberculosis* (Yp), or EPEC strain E2348/69. Unlike *Y. pseudotuberculosis* or other EPEC strains, E2348/69 harbors a defective *cif* locus. Species specific expression plasmids were used to complement *cif* mutants. Immunoprecipitates were then separated by SDS-PAGE and probed with the indicated antibodies. Cif$_{Yp}$-C117A-FLAG denotes a point mutation within the enzyme's catalytic triad, conserved amongst Cif proteins, that is essential for deamidation of NEDD8. The inability of Cif$_{Yp}$-C117A-FLAG to block ubiquitylation of Perforin-2 is not due to a lack of expression (**Figure 5—figure supplement 1**).

The following figure supplement is available for figure 5:

**Figure supplement 1**. Cif$_{Yp}$ expression.

ubiquitylation of Perforin-2. Likewise, an EPEC strain (E2348/69) carrying a naturally defective *cif* allele was unable to block ubiquitylation of Perforin-2 unless it was transformed with a Cif expression plasmid (*Marches et al., 2003*) (*Figure 5*).

Structural and mutagenic studies have shown that the enzymatic activity of Cifs is dependent upon a catalytic triad composed of Cys, His, and Gln residues (*Hsu et al., 2008*; *Crow et al., 2009*; *Jubelin et al., 2009*, *2010*; *Yao et al., 2009*; *Cui et al., 2010*). Therefore, to exclude the possibility that the inhibition of Perforin-2 ubiquitylation is independent of NEDD8 deamidation we used site directed mutagenesis to change the codon of Cys117, which along with His173 and Gln193 forms the catalytic triad to Cif$_{Yp}$, to an Ala codon. In contrast to the Cif$_{Yp}$-FLAG expression plasmid, the Cif$_{Yp}$-C117A-FLAG expression plasmid was unable to complement a *cif::aadA* mutation in *Y. pseudotuberculosis* even though the two proteins were expressed equally well (*Figure 5* and *Figure 5—figure supplement 1*). Thus, these results demonstrate that the deamidase activity of Cif is intrinsic to its ability to block ubiquitylation of Perforin-2. In addition, these results provide further evidence that Perforin-2 is ubiquitylated by a CRL complex because deamidation of NEDD8 inhibits CRL activity.

## Cif promotes pathogen survival in vitro and in vivo

Having found that Cif blocks ubiquitylation of Perforin-2 and that ubiquitylation is essential to Perforin-2 activity, we next evaluated the biological relevance of these findings using in vitro and in vivo infection models. We found that transformation of a Cif⁻ strain with a Cif$_{Ec}$ expression plasmid resulted in an EPEC strain that was able to block Perforin-2-dependent killing by stimulated Caco-2 cells (*Figure 6A*). In addition, Cif blocked killing as effectively as Perforin-2 siRNA (*Figure 6A*). In contrast, the *cif* mutant was rapidly killed unless expression of Perforin-2 was ablated with siRNA (*Figure 6B*). Similar results were obtained with a murine mucosal epithelial cell line infected with wild-type *Y. pseudotuberculosis* and an isogenic *cif::aadA* mutant (*Figure 6C,D*). Transformation of

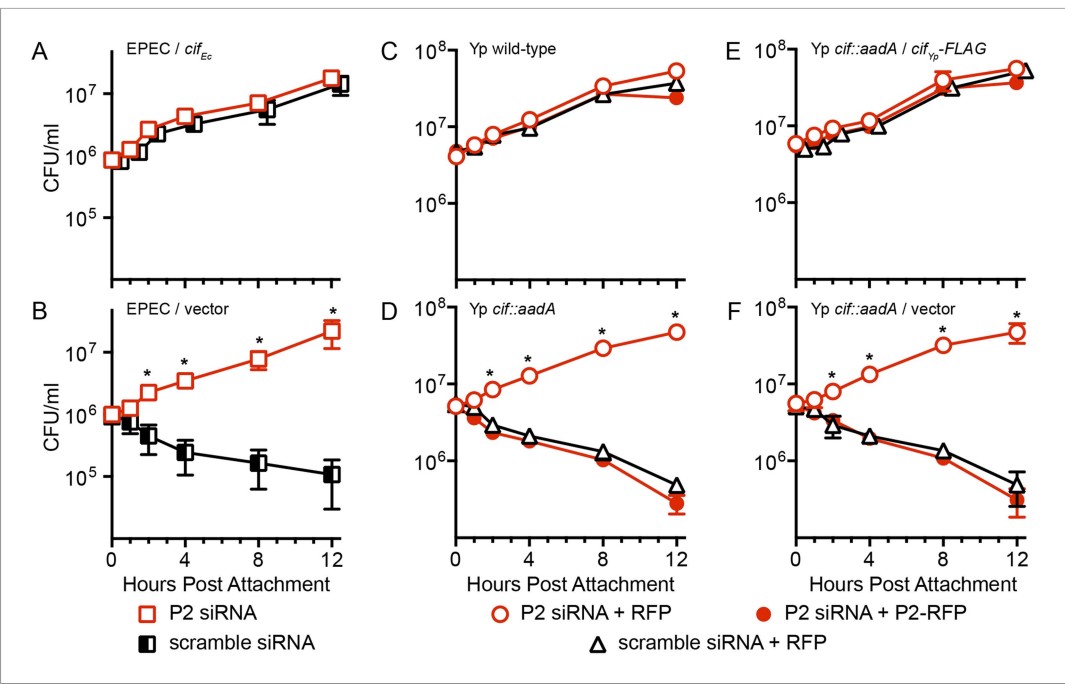

**Figure 6**. The bacterial effector protein Cif blocks the bactericidal activity of Perforin-2. Caco-2 cells, a human intestinal epithelial cell line, were transfected with Perforin-2 specific or scramble siRNAs. The cells were subsequently infected with EPEC strain E2348/69, which carries a naturally disrupted *cif* gene, transformed with a (**A**) Cif$_{Ec}$ expression plasmid or (**B**) vector control. Alternatively CMT93 cells, a murine epithelial cell line, were cotransfected with the indicated siRNA and RFP or Perforin-2-RFP expression plasmids. Transfected cells were subsequently infected with (**C**) wild-type *Y. pseudotuberculosis*, (**D**) a *cif* mutant, (**E**) the mutant transformed with a Cif$_{Yp}$-FLAG expression plasmid, or (**F**) vector control. All mammalian cells were activated with IFN-γ for 24 hr prior to infection. *p < 0.05 by Student's *t*-test, *n* ≥ 3.

the *cif::aadA* mutant with a Cif$_{yp}$-FLAG expression plasmid restored resistance to Perforin-2 while a vector control plasmid did not (*Figure 6E,F*). In addition, the specificity of Perforin-2 knockdown was confirmed by transfection of siRNA-treated cells with a Perforin-2-RFP expression plasmid lacking the siRNA targeting region. As expected, Cif$^-$ bacteria were killed by siRNA ablated cells that express Perforin-2-RFP but not RFP (*Figure 6C–F*). We also evaluated the impact of a point mutation within the catalytic triad of Cif$_{Yp}$. Consistent with the inability of Cif$_{Yp}$-C117A-FLAG to block ubiquitylation of Perforin-2, we found that it was also unable to block Perforin-2-dependent killing (*Figure 7*).

We next evaluated the role of Cif in vivo by using *Y. pseudotuberculosis* in a murine infection model (*Figure 8*). This human and rodent pathogen colonizes the distal ileum and proximal colon. Subsequent invasion of the underlying lymphatic tissue facilitates *Y. pseudotuberculosis* dissemination to the spleen and liver followed by death several days after inoculation (*Marra and Isberg, 1997*; *Mecsas et al., 2001*; *Logsdon and Mecsas, 2003*). In our studies we found that 80% of C57Bl/6 mice perished 6–10 days after orogastric inoculation with $10^8$ CFU of wild-type *Y. pseudotuberculosis*. These results are consistent with other studies that used the same inoculation method and similar infectious doses (*Mecsas et al., 2001*; *Logsdon and Mecsas, 2003*). In sharp contrast to wild-type *Y. pseudotuberculosis*, all of the animals that were inoculated with the Cif$^-$ mutant survived even though they were inoculated at the same infectious dose (*Figure 8*). Thus, these results demonstrate that Cif is an important virulence factor of *Y. pseudotuberculosis* and by extension, EPEC and other Cif$^+$ pathogens.

To determine if the effects of Cif in vivo are primarily through disruption of Perforin-2 bactericidal activity or another pathway(s), we inoculated C57Bl/6 × 129X1/SvJ *Mpeg1* +/+, +/− and −/− mice with $10^6$ CFU of Cif$^+$ and Cif$^-$ *Y. pseudotuberculosis*. We note that these orogastric inocula are considerably lower than the ~$10^9$ CFU that are typically used because we had previously determined that Perforin-2 deficient mice are hypersensitive to infectious agents (*Mecsas et al., 2001*; *Logsdon and Mecsas, 2003*). As expected with such a low dose, wild-type mice survived equally well when infected with wild-type or mutant bacteria (*Figure 9A*). Although no wild-type mice perished with this low infectious dose, some Cif$^+$ bacteria were recovered from the spleens of infected animals (*Figure 10A*). In contrast, Cif$^-$ bacteria did not disseminate to the spleens of wild-type mice. In *Mpeg1* heterozygotes, Cif$^+$ bacteria were clearly more virulent than Cif$^-$ bacteria and significantly more Cif$^+$ than Cif$^-$ bacteria were recovered from the organs and blood of infected animals (*Figures 9B, 10B*). Statistically insignificant differences were observed between Cif$^+$ and Cif$^-$ bacteria in *Mpeg1* knockout mice (*Figures 9C, 10C*). However, a gene dosage effect is evident when organ loads are compared across murine genotypes (*Figure 10*). In this comparison there is a consistent trend of *Mpeg1* −/− mice having the highest organ loads, wild-type mice the lowest, and intermediate loads in *Mpeg1* heterozygotes.

To confirm our observations with C57Bl/6 × 129X1/SvJ mice we conducted similar experiments with 129X1/SvJ *Mpeg1* +/+, +/−, and −/− mice. As with C57Bl/6 × 129X1/SvJ mice, there was not a significant difference between Cif$^+$ and Cif$^-$ *Y. pseudotuberculosis* in wild-type 129X1/SvJ mice; although, a few animals died in both cases (*Figure 9D*). In contrast, nearly all of the *Mpeg1* knockout mice died (*Figure 9F*). As with C57Bl/6 × 129X1/SvJ mice, there was not a significant difference between Cif$^+$ and Cif$^-$ bacteria in 129X1/SvJ knockout mice. However a difference with *Mpeg1* heterozygotes was

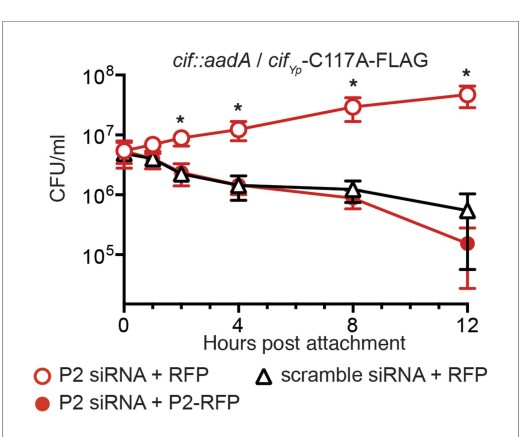

**Figure 7**. Enzymatically inactive Cif cannot block Perforin-2-dependent killing. CMT93 cells were transfected with the indicated siRNAs and RFP or Perforin-2-RFP expression plasmids. Following stimulation with IFN-γ, transfected cells were infected with a *Y. pseudotuberculosis cif::aadA* mutant transformed with a Cif$_{Yp}$-C117A-FLAG expression plasmid. The C117A point mutation within the enzyme's conserved catalytic triad abolishes its ability to block ubiquitylation of Perforin-2. *Statistically significant (p < 0.05) difference between Perforin-2 siRNA + RFP and scramble siRNA + RFP or Perforin-2 siRNA + Perforin-2-RFP by one-way ANOVA with Bonferroni multiple-comparisons post-hoc test; $n \geq 3$.

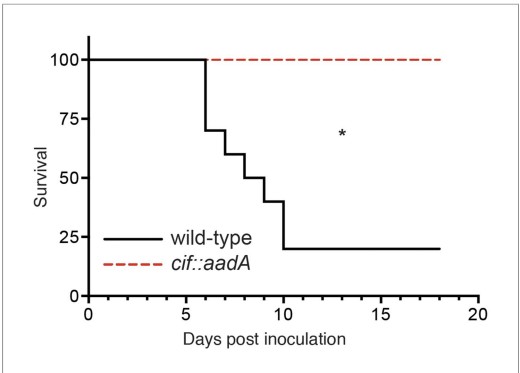

**Figure 8**. Cif enhances pathogenicity in vivo. Survival curves of C57Bl/6 mice inoculated orogastrically with $10^8$ CFU of wild-type *Y. pseudotuberculosis* or an isogenic *cif::aadA* mutant. Animals were weighed daily and euthanized if weight loss exceeded 20%. *p < 0.05 by log-rank (Mantel–Cox) test, *n* = 9–10 mice per group.

again observed (*Figure 9E*). In this latter group Cif⁺ bacteria killed 84% of the heterozygotes while Cif⁻ bacteria killed none. Because our results with 129X1/SvJ mice replicate those obtained with C57Bl/6 × 129X1/SvJ mice, we conclude that Perforin-2 deficiency is the primary cause of increased sensitivity to bacterial pathogens. Our results also demonstrate that Cif enhances pathogenicity primarily through the disruption of Perforin-2 activity and not through other pathways, such as cell cycling, that are known to be disrupted by Cif. The molecular mechanism of this effect is undoubtedly the blockade of NEDD8-dependent ubiquitylation of Perforin-2 by Cif, as suggested by our in vitro studies.

## Monoubiquitylation of Perforin-2 signals its cellular redistribution

Having determined that ubiquitylation is necessary for Perforin-2 activity, we next sought to determine the number of and linkages between ubiquitin monomers as these properties determine the fate of ubiquitylated proteins. For example, polyubiquitylation through K48 linkages target proteins for degradation by the proteasome while K63 polyubiquitylation is often involved in cell signaling. Alternatively, ubiquitylation may terminate after the addition of a single ubiquitin molecule to the target protein. This is termed monoubiquitylation and typically serves as a sorting signal that directs the modified protein to one or more subcellular compartments. To determine the type of Perforin-2 ubiquitylation we probed Perforin-2-GFP immunoprecipitates with a linkage independent ubiquitin antibody (P4D1) as well as antibodies specific for K48 and K63 polyubiquitylation (*Figure 11*). Consistent with previous results, the P4D1 antibody recognized Perforin-2-GFP immunoprecipitated from transfected knockout MEFs that were stimulated with IFN-γ and LPS. Ubiquitylation was not detected in the absence of LPS. In addition, MLN4924, a small molecule that inhibits cullin neddylation, also blocked LPS stimulated ubiquitylation of Perforin-2-GFP (*Figure 11*) (*Soucy et al., 2009*). This latter result provides additional confirmation that Perforin-2 is ubiquitylated by a CRL complex. We estimate that ubiquitylation increases the mass of Perforin-2-GFP by ≤10 kDa (*Figure 11—figure supplement 1*). Given that ubiquitin has a mass of 8.5 kDa, this suggest that Perforin-2-GFP is monoubiquitylated. Since Perforin-2-GFP is a 105 kDa molecule, this relatively small increase in its mass also explains why mass shifts are not apparent on gels of less resolution; for example *Figure 2B*. Consistent with monoubiquitylation, antibodies specific for K48 or K63 polyubiquitylation did not recognize ubiquitylated Perforin-2-GFP (data not shown). Although other forms of polyubiquitylation through K6, K11, K27, K29, K33, or M1 linkages—in addition to chains of mixed linkages—are also possible, polyubiquitylation is inconsistent with the small increase in the mass of Perforin-2-GFP. This suggests monoubiquitylation serves as a sorting signal that directs Perforin-2 to a specific compartment within the cell or location on the plasma membrane.

To determine if ubiquitylated Perforin-2 has a cellular distribution different than non-ubiquitylated Perforin-2 we transfected *Mpeg1* knockout MEFs with Perforin-2-RFP and Perforin-2-KQ-RFP expression plasmids. Due to site directed mutagenesis of the three most highly conserved lysines within the carboxy-terminal tail of Perforin-2 the latter fusion protein cannot be ubiquitylated (*Figure 4*). Perforin-2-RFP was observed in distinct punctate bodies following stimulation of the transfected cells with IFN-γ and LPS (*Figure 12A*). Three dimensional projections of the acquired confocal Z-stacks revealed the vesicular structure of the punctate bodies (*Video 1*). This is in sharp contrast to Perforin-2-KQ-RFP which had a diffuse, perinuclear distribution in LPS stimulated cells (*Figure 12B* and *Video 2*). Since we have already shown that LPS causes ubiquitylation of Perforin-2 and K-to-Q mutations in its cytosolic tail abolish ubiquitylation, these results suggest the subcellular distribution of Perforin-2 is determined by monoubiquitylation.

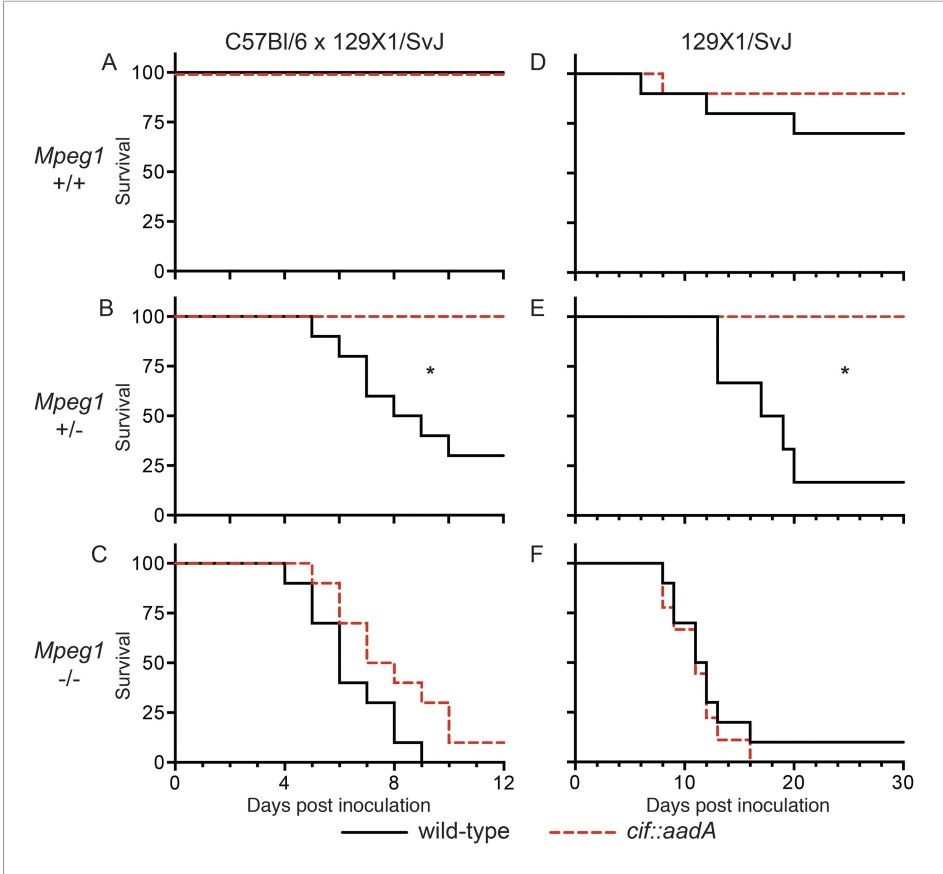

**Figure 9**. Bacterial inhibitors of neddylation abolish the bactericidal activity of Perforin-2 in vivo. Survival curves for two different lineages of *Mpeg1* +/+, +/−, and −/− mice following orogastric inoculation with $10^6$ CFU of wild-type *Y. pseudotuberculosis* or an isogenic *cif::aadA* mutant. (**A–C**) $n$ = 6–10 mice per group. (**D–F**) $n$ = 22–28 mice per group. *$p < 0.05$ by log-rank (Mantel–Cox) test.

To further test the hypothesis that ubiquitin serves as a sorting signal for Perforin-2 in response to PAMPs we infected the transfected MEFs with Cif+ and Cif− *Y. pseudotuberculosis* (*Figure 13*). When the MEFs were infected with wild-type bacteria that block ubiquitylation of Perforin-2, the distribution of Perforin-2-RFP was diffuse and perinuclear (*Figure 13A*). In contrast, Perforin-2-RFP was located in punctate bodies when the cells were infected with Cif− bacteria that are incapable of blocking ubiquitylation of Perforin-2 (*Figure 13B*). To further test our hypothesis we also infected MEFs expressing Perforin-2-KQ-RFP. As expected, Perforin-2-KQ-RFP remained diffuse and perinuclear when the cells were infected with either Cif− or Cif+ bacteria (*Figure 13C,D*). These results were not restricted to knockout MEFs because similar results were obtained with transfected CMT93 cells (*Figure 13—figure supplements 1, 2*). Thus we conclude that CRL-dependent monoubiquitylation of Perforin-2 initiates a journey that ultimately terminates with the insertion and polymerization of Perforin-2 in a bacterial membrane.

## Discussion

In this study we have shown that Perforin-2 is monoubiquitylated by a CRL complex in response to extracellular bacteria or PAMPs and that ubiquitylation is required for Perforin-2's bactericidal activity against both extracellular, cell adherent, and intracellular bacteria. Although we initially identified an interaction between UBC12 and Peforin-2 in a yeast two-hybrid screen, this result is rather curious as this NEDD8 conjugating enzyme is not thought to directly interact with CRL substrates. Thus, it is not clear if our initial two-hybrid hit reflects a bona fide or fortuitous interaction. Nevertheless, we subsequently demonstrated that Perforin-2 coimmunoprecipitates CUL1, UBC12, and βTrCP from

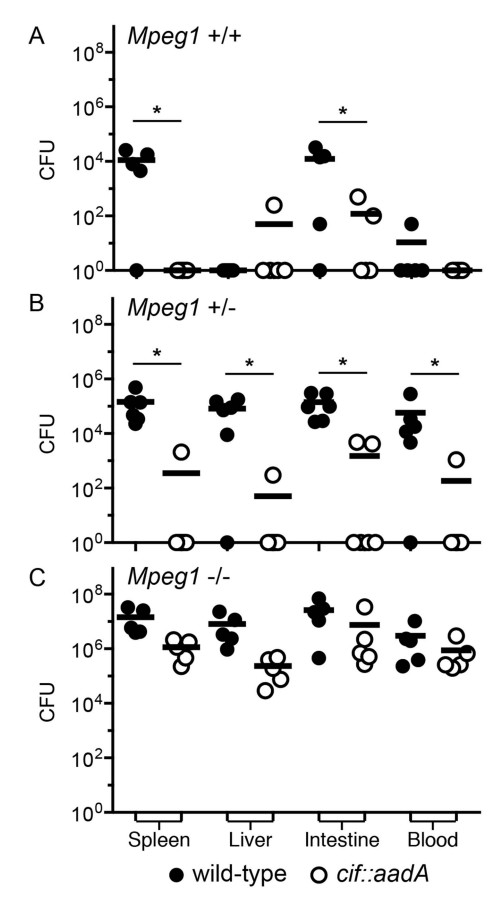

**Figure 10**. Cif diminishes the bactericidal activity of Perforin-2 in vivo. Representative organ loads of C57Bl/6 × 129X1/SvJ *Mpeg1* (**A**) +/+, (**B**) +/− and (**C**) −/− mice infected with Cif⁺ or Cif⁻ *Y. pseudotuberculosis*. Animals were sacrificed 10 days after orogastric inoculation with $10^6$ CFU. Samples were normalized by organ weight and each experiment was repeated twice. Horizontal lines represent the mean. Statistical analysis was performed by the nonparametric Kolmogorov–Smirnov test with Dunn's multiple comparison test; *$p < 0.05$, $n \geq 5$. DOI: 10.7554/eLife.06505.013

lysates of mammalian cells. As an F-box substrate receptor, βTrCP may provide the primary and specific interaction between Perforin-2 and the CRL (*Figure 14*). It is also likely that phosphorylation of Perforin-2 precedes its ubiquitylation because βTrCP has been shown to recognize phosphorylated substrates (*Winston et al., 1999*; *Strack et al., 2000*; *Wu et al., 2003*). However we have not yet determined whether or not this is the case for Perforin-2.

We have demonstrated that siRNA knockdown of either CUL1 or UBC12 blocked ubiquitylation of Perforin-2 and Perforin-2-dependent killing of extracellular bacteria. These latter results confirm the biological relevance of the coimmunoprecipitates. We have also shown that MLN4924, a small molecule inhibitor of the NEDD8 activating enzyme also blocks ubiquitylation of Perforin-2 (*Soucy et al., 2009*) (*Figure 11*). Thus, we conclude that PAMPs trigger CRL-dependent monoubiquitylation of Perforin-2's cytosolic domain. As described below, this conclusion is further supported by our studies with the bacterial effector protein Cif. Mutagenesis of conserved lysines within Perforin-2's carboxy-terminus also prevented its ubiquitylation and abolished its bactericidal activity. We have also shown that these mutations prevent the cellular redistribution of Perforin-2 in response to LPS. Likewise, we have shown that Cif blocks Perforin-2 trafficking. Because the destruction of bacteria involves a direct interaction between Perforin-2 and a bacterium, the inability of a cell to route Perforin-2 abolishes its bactericidal activity (*McCormack et al., 2015*).

Perforin-2 was originally identified as a potential marker of mature macrophages (*Spilsbury et al., 1995*). However its expression is not limited to professional phagocytes. In human and murine cells, including non-immune cells, the expression of Perforin-2 can be induced by interferons or intracellular bacteria (*Fields et al., 2013*; *McCormack et al., 2013b*). The gene encoding Perforin-2 is evolutionarily ancient and is present in all but one species of bony vertebrates (*D'Angelo et al., 2012*). As with mammalian Perforin-2, infectious bacteria also induce the expression of a Perforin-2 ortholog (MPEG1.2) in zebrafish. Although ubiquitylation was not investigated, upregulation of MPEG1.2 has been shown to require NFκB and the TLR adaptor MyD88 (*Benard et al., 2014*). In non-vertebrate species the expression of Perforin-2 orthologs is constitutive in some tissues and/or inducible by pathogenic bacteria or LPS (*He et al., 2011*; *Kemp and Coyne, 2011*; *Bathige et al., 2014*). In addition to similar patterns of expression, the function of Perforin-2 is also conserved across species. For example, Perforin-2 has been shown to limit the burden of *Mycobacterium marinum* and *S. typhimurium* in zebrafish embryos (*Benard et al., 2014*). Additional studies indicate that Perforin-2 orthologs provide similar bactericidal activity in invertebrates (*He et al., 2011*; *Kemp and Coyne, 2011*; *Bathige et al., 2014*).

Because Perforin-2 has broad bactericidal activity, is expressed in a variety of cell types and tissues, and is evolutionarily ancient, it is likely that evolution has endowed bacterial pathogens with novel

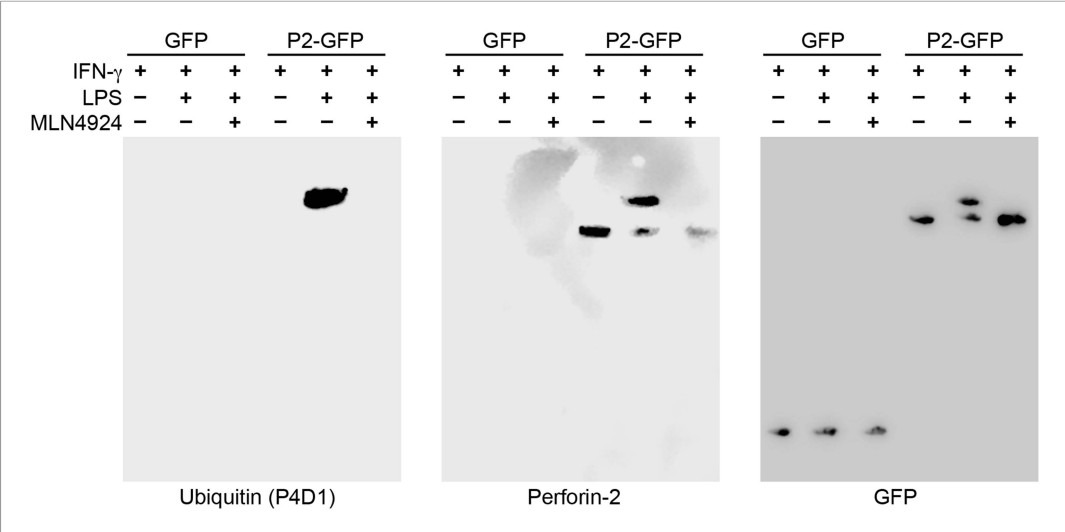

**Figure 11**. Perforin-2 is monoubiquitylated. *Mpeg1* knockout MEFs were transfected with GFP or Perforin-2-GFP expression plasmids and stimulated with IFN-γ in the presence or absence of LPS. Some cultures were also treated with MLN4924, a small molecule inhibitor of neddylation. The resulting GFP immunoprecipitates were probed with the indicated antibodies in Western blots. Consistent with previous results ubiquitylation of Perforin-2 was dependent upon LPS and neddylation when the immunoprecipitates were probed with a non-linkage specific ubiquitin antibody. When the same immunoprecipitates were probed with antibodies specific for K48 or K63 linkages, ubiquitylation was not detected. From additional Western blots we estimated that ubiquitylation increases the mass of Perforin-2-GFP by ~10 kDa (*Figure 11—figure supplement 1*). These results indicate the Perforin-2 is a monoubiquitylated protein.

The following figure supplement is available for figure 11:

**Figure supplement 1**. Monoubiquitylation of Perforin-2.

strategies to defeat Perforin-2. In this study we have identified the bacterial effector protein Cif as one such example (*Figure 14C*). We have shown that Cif blocks ubiquitylation and trafficking of Perforin-2. This blockade prevents Perforin-2 from reaching the bacterium and explains how Cif inhibits Perforin-2-dependent killing in vitro and enhances pathogenicity in vivo. These properties are abolished by a mutation in the catalytic site of Cif$_{Yp}$. An equivalent mutation has been previously shown to abolish the deamidase activity of Cif$_{Ec}$ (*Marches et al., 2003*; *Hsu et al., 2008*; *Jubelin et al., 2009*). Because we have also demonstrated that the bactericidal activity of Perforin-2 is dependent upon its ubiquitylation by a CRL complex, we can conclude that Cif abolishes the bactericidal activity of Perforin-2 through deamidation of NEDD8 (*Figure 14C*). Moreover, the absence of Perforin-2 in *Mpeg1* knockout mice negated the benefit of Cif in vivo. Similar results were observed by siRNA knockdown of Perforin-2 expression in vitro. These results indicate that the inactivation of Perforin-2 is not a secondary or ancillary function of Cif. Rather, our results demonstrate that Cif enhances virulence primarily through inactivation of Perforin-2 as opposed to other mechanisms such as reduced turn-over of epithelial cells. Although Cif is the first definitive example of a bacterial product that blocks the bactericidal activity of Perforin-2, it is likely that others remain to be discovered.

Our observation that Cif blocks monoubiquitylation of Perforin-2 and its trafficking within the cell is consistent with other transmembrane proteins for which monoubiquitylation is known to serve as a sorting signal directing their subcellular localization (*Hicke and Dunn, 2003*; *MacGurn et al., 2012*). In our accompanying manuscript we provide evidence that suggest Perforin-2 polymerizes and forms pores on bacteria adherent to the plasma membrane or within the endosomal lumen (*McCormack et al., 2015*). Thus, we propose a model in which monoubiquitylation of Perforin-2 initiates its trafficking towards bacteria at either location to facilitate Perforin-2-dependent destruction of bacterial pathogens. We also observed that the MACPF and P2 domains of Perforin-2 are cleaved from its transmembrane and cytosolic domains (*McCormack et al., 2015*). Although it is not yet known when

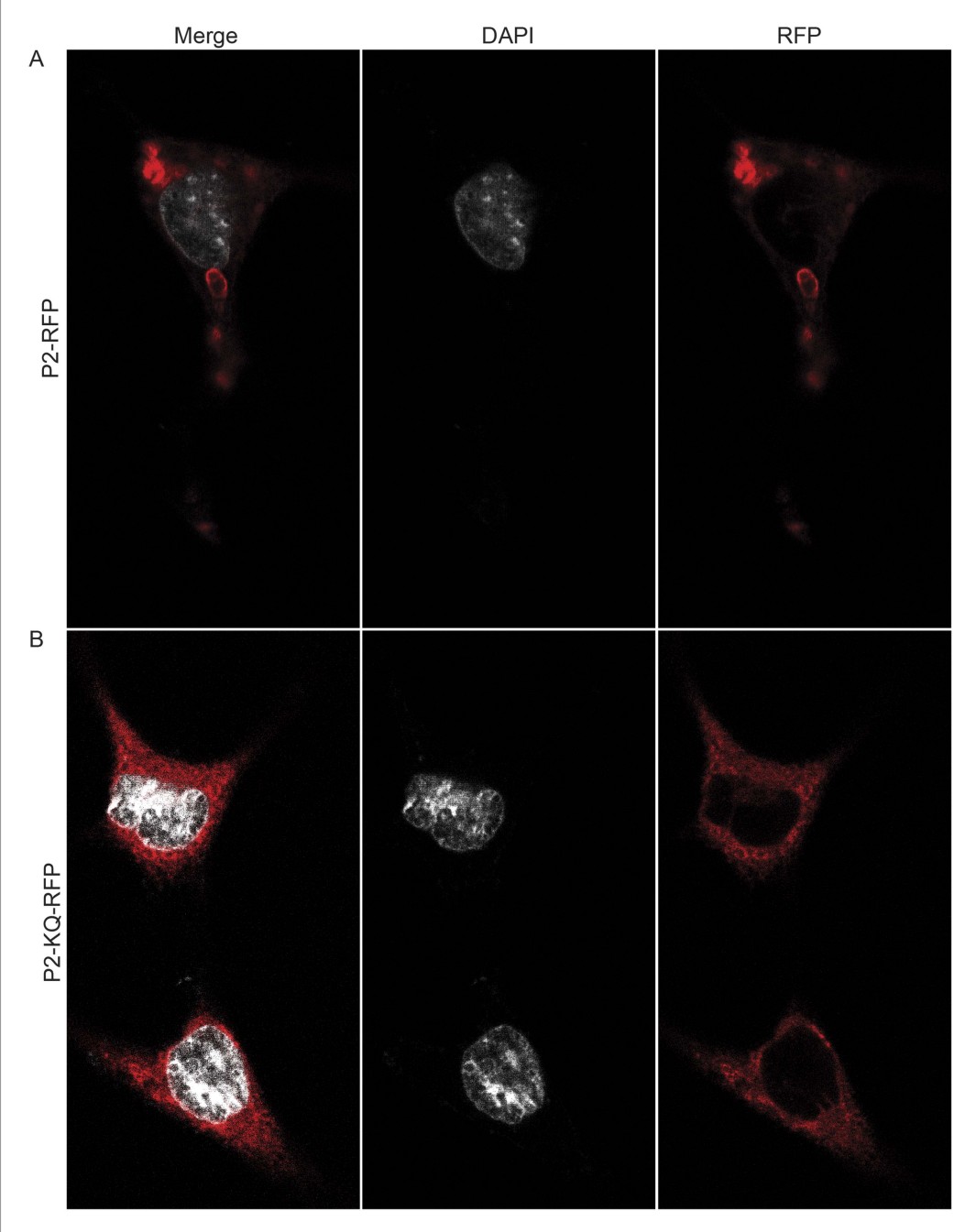

**Figure 12**. Ubiquitylation determines the subcellular distribution of Perforin-2. *Mpeg1* knockout MEFs were transfected with (**A**) Perforin-2-RFP or (**B**) Perforin-2-KQ-RFP expression plasmids, the latter of which carries K-to-Q mutations of conserved lysines in the cytosolic tail of Perforin-2 that abolish ubiquitylation. Transfected cells were stimulated with IFN-γ for 24 hr prior to addition of LPS. Cells were fixed within 15 min of LPS addition and counter stained with DAPI. Images were acquired on a Leica confocal microscope with a 63× objective.

cleavage occurs within the Perforin-2-dependent killing pathway, cleavage may explain the lack of colocalization of Perforin-2-RFP and Cif⁻ bacteria in confocal microscopy (*Figure 13*). Although bacterial pathogens such as *Y. pseudotuberculosis* and EPEC block the critical event—ubiquitylation—that initiates the trafficking of Perforin-2 to its final destination, other pathogens may target events further upstream or downstream of ubiquitylation in order to preserve their cellular integrity and flourish.

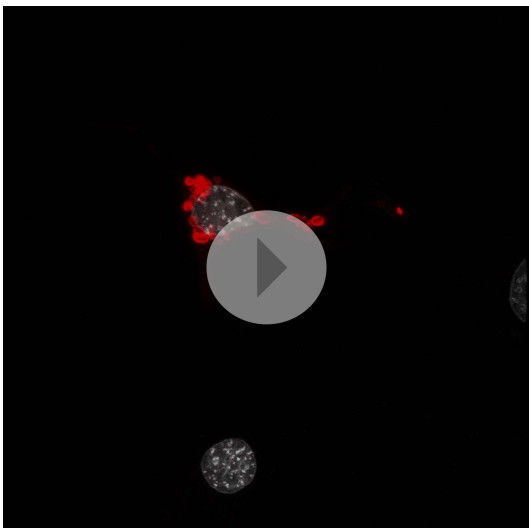

**Video 1.** Perforin-2-RFP is located in vesicular bodies following stimulation with LPS. *Mpeg1* knockout MEFs were transfected with a Perforin-2-RFP expression plasmid and stimulated overnight with IFN-γ. After stimulation with LPS for 15 min, cells were fixed and counter stained with DAPI. Three dimensional projections of the acquired confocal Z-stacks reveal that Perforin-2 is located in punctate bodies with vesicular structure in LPS stimulated cells.

**Video 2.** Redistribution of Perforin-2 is dependent upon ubiquitylation. *Mpeg1* knockout MEFs were transfected with a Perforin-2-KQ-RFP expression plasmid. This fusion protein cannot be ubiquitylated because it carries three K-to-Q mutations in the carboxy-terminal tail of Perforin-2. Transfected cells were stimulated overnight with IFN-γ then LPS. Cells were fixed 15 min after exposure to LPS and counter stained with DAPI. Three dimensional projections of the acquired confocal Z-stacks reveal that Perforin-2-KQ-RFP has a diffuse, perinuclear distribution in LPS stimulated cells.

## Materials and methods

### Cell culture and bacterial organisms

RAW 264.7 (TIB-71), CMT93 (CCL-223), Caco-2 (HTB-37) cell lines were obtained from American Type Culture Collection, Manassas, VA. MEFs and murine PMN were isolated as previously described (*Luo and Dorf, 2001*; *Scheuner et al., 2001*). All cells were cultured at 37°C in a humidified atmosphere containing 5% $CO_2$ following ATCC recommendations for culture conditions. *S. typhimurium* SL1344 (gift from Dr J Galán, Yale University) and EPEC strain E2348/69 were cultured in Luria–Bertani broth (LB) at 37°C. *Y. pseudotuberculosis* YPIII pIB102 (*Bolin and Wolf-Watz, 1984*) was cultured in HIB at 27°C, or HIB plus 2.5 mM $CaCl_2$ at 37°C when subculturing. Human and murine IFN-γ was purchased from Peprotech (Rocky Hill, NJ, United States), LPS purchased from Invivogen (San Diego, CA, United States), and MLN4924 was purchased from Active Biochem (Maplewood, NJ, United States) and EMD Millipore (Billerica, MA, United States).

Because EPEC is unable to translocate Cif of *Y. pseudotuberculosis* (*Jubelin et al., 2009*), *cif* mutants were complemented with alleles cloned from their respective species. Accordingly, EPEC strain E2348/69, a spontaneous Cif⁻ mutant, was transformed with the $Cif_{Ec}$ expression plasmid pCif$_{wt}$ or vector control pBRSK (gift from Drs E Oswald and F Taieb, University of Toulouse, France) (*Marches et al., 2003*; *Oswald et al., 2005*). Plasmid pCif$_{YP}$-FLAG expresses Cif$_{Yp}$-FLAG. It was constructed by PCR amplification of *cif*$_{Yp}$ from YPIII with primers 1118 (ATGAAGCTTAGCCCTAATACCATTAGTCC) and 1119 (TCTGGTACCATTACAGTGAGTTTTAATG). Underlined sequences indicate primer-template mismatches. The PCR product was then digested with HindIII and KpnI then ligated into the same sites of pFLAG-CTC (Sigma-Aldrich, St. Louis, MO, United States). Oligonucleotide directed mutagenesis of pCifYP-FLAG with primers 1136 (GCGGGTGTCACGGCAAATACC) and 1137 (AACGGGTTCTATTATGCGC) was used to construct pCifYP-CA-FLAG which expresses Cif$_{Yp}$-C109A-FLAG. Strain GPM1769 (*cif::aadA*) was constructed by lambdaRED-mediated recombination in a multistep process. First, *aadA* was amplified from pHKLac1 with primers 238 (CCTGGCAGTT-TATGGCGGGCGT) and 1131 (GCGCATGCTGATCTTCAGATCCTC). The PCR product was then

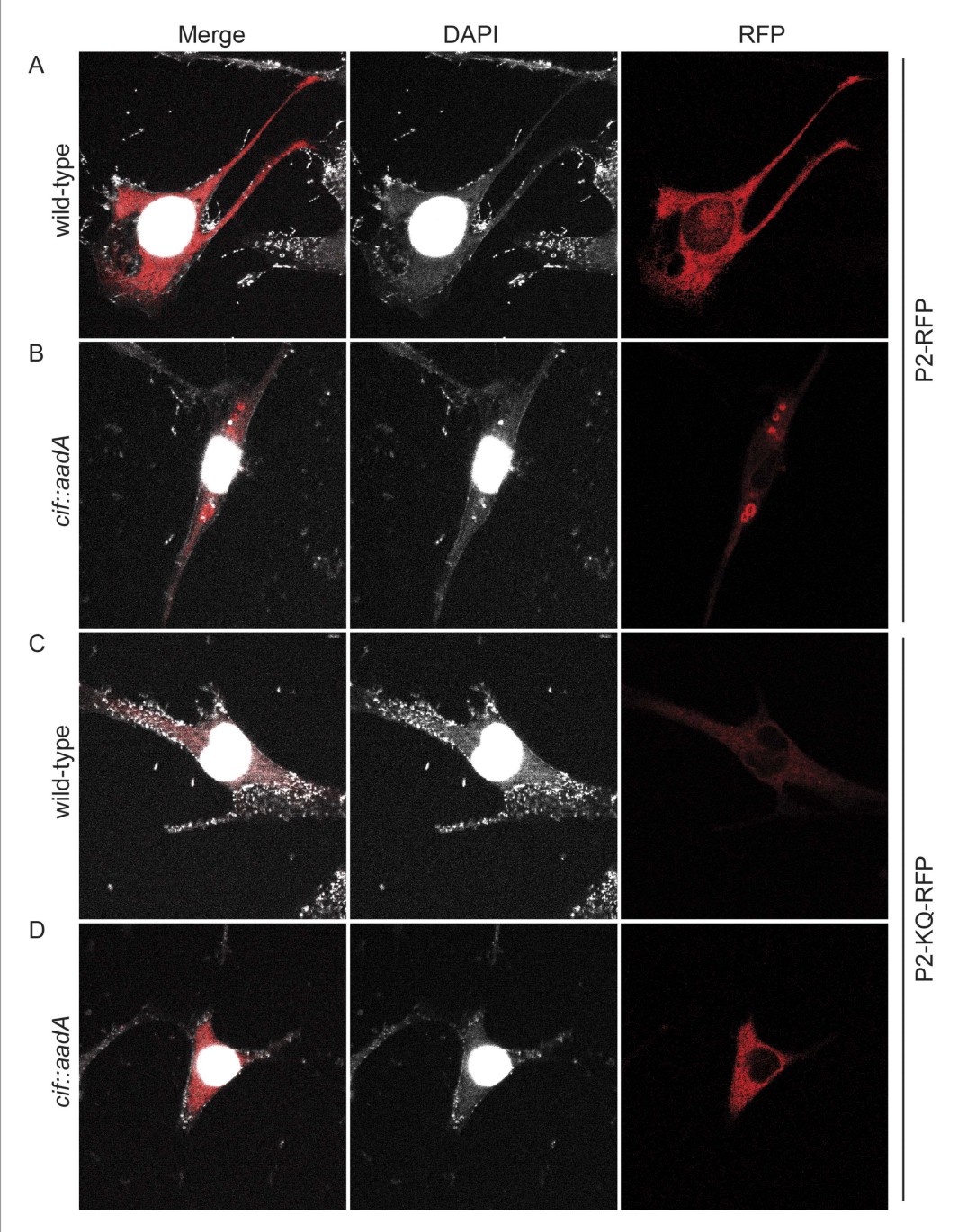

**Figure 13**. Cif blocks trafficking of Perforin-2. *Mpeg1* knockout MEFs were transfected with Perforin-2-RFP or Perforin-2-KQ-RFP expression plasmids and stimulated with IFN-γ for 24 hr prior to infection with (**A**, **B**) wild-type *Y. pseudotuberculosis* or (**C**, **D**) an isogenic *cif::aadA* mutant. Cells were fixed 15 min after infection and counter stained with DAPI. Similar results were obtained when transfected CMT93 cells were infected with Cif+ and Cif− bacteria expressing GFP (*Figure 13—figure supplements 1, 2*). Images were acquired on a Leica confocal microscope with a 63× objective.

The following figure supplements are available for figure 13:

**Figure supplement 1**. Perforin-2 relocalizes from perinuclear to punctate bodies upon ubiquitylation.

**Figure supplement 2**. Perforin-2 perinuclear localization in noninfected CMT93s.

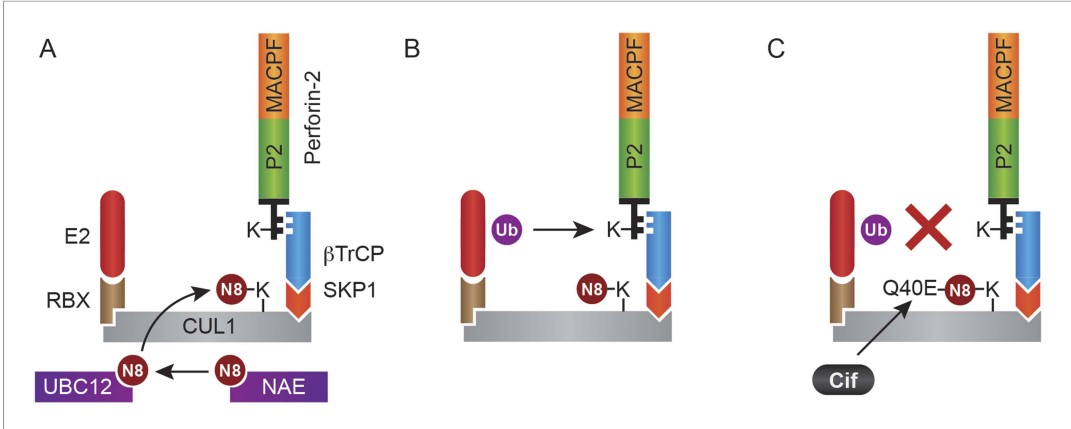

**Figure 14**. Monoubiquitylation of Perforin-2 and inhibition by Cif. (**A**) CRLs are modular complexes assembled upon cullin scaffolds such as CUL1. F-box proteins such as βTrCP, which we have shown interacts with Perforin-2, provide substrate specificity within the multisubunit complex. Because βTrCP is known to recognize phosphorylated substrates, it is likely that TLR signaling activates one or more kinases that phosphorylate Perforin-2 prior to its association with the CRL. SKP1 serves as an adaptor between βTrCP and the cullin scaffold. An ubiquitin E2 ligase and a RING protein—either RBX1 or 2—dock at the opposite end of the elongated cullin. The E1 enzyme NEDD8 activating enzyme (NAE) activates NEDD8. The E2 ligase UBC12 then covalently binds NEDD8 to CUL1. (**B**) Neddylation of CUL1 activates the CRL complex and is essential for subsequent monoubiquitylation of Perforin-2. (**C**) The deamidation of NEDD8 by the bacterial effector protein Cif inhibits ubiquitylation of Perforin-2 and its subsequent trafficking to and destruction of extracellular bacteria at the plasma membrane or intracellular bacteria within an endosomal compartment. Ub, ubiquitin; N8, NEDD8; N8-Q40E, deamidated NEDD8.

digested with ClaI and SphI then ligated into the BstBI and SphI sites within $cif_{Yp}$ of pCifYP-FLAG. The cassette and $cif_{Yp}$ flanking sequences were then amplified with primers 1133 (GCCACCCTAAGTG-CACG) and 1134 (GAGATGATCTGCACGCAG). Finally, the PCR product was recombined into YPIII/pSIM6 as previously described (*Datta et al., 2006*). Recombinants were isolated by selection for simultaneous resistance to spectinomycin and streptomycin, and confirmed by PCR with primers 1131 (<u>GCGCA</u>TGCTGATCTTCAGATCCTC) and 1132 (<u>GACGGA</u>TCCATGATAGCCTGAGCCAG).

## Generation of *Mpeg1* knockout mice

For the generation of *Mpeg1* knockout mice the targeting vector was linearized and electroporated into RW-4 ES cells originating from the 129X1/SvJ strain, followed by selection in G418. Targeted clones were screened by PCR. From 90 clones, 2 positive clones were selected that had undergone homologous recombination and were identified through Southern blot analysis. One ES clone was utilized for the generation of chimeric mice by injection using C57Bl/6J blastocysts as the host. The resulting female chimeras were further mated with C57Bl/6J male mice for germ line transmission. The heterozygous mice ($F_1$ mice) were interbred to obtain wild-type, heterozygous, and homozygous littermates ($F_2$). C57Bl/6 × 129X1/SvJ animals utilized in these experiments were backcrossed between 7–10 times for these experiments. Mouse genotype was determined by PCR using PCR probes MP10 and MP11.

To generate 129 pure animals without potential passenger mutations, chimeric mice were mated with 129X1/SvJ animals, and assessed for germ line transmission. The heterozygous mice were then inbred to obtain a genetically pure 129X1/SvJ strain. Mouse genotype was determined by PCR utilizing PCR probes MP10 and MP11.

Animals were bred at the University of Miami, Miller School of Medicine Transgenic Core Facility. Mice were allowed to freely access food and water and were housed at an ambient temperature of 23°C on a 12 hr light/dark cycle under specific pathogen-free condition. Animal care and handling were performed as per IACUC guidelines.

## *Y. pseudotuberculosis* in vivo infection

All animal experiments were performed in accordance with the University of Miami Animal Care and Use Committee guidelines. The animal genotype was blinded prior to the experiment to limit bias.

C57Bl/6 mice were infected orogastrically with $10^6$ colony-forming units of *Y. pseudotuberculosis* as previously described (*Schweer et al., 2013*). Mice were weighed daily throughout the experiment; the animals were euthanized after greater than 20% weight loss. Animal bacterial colonization was confirmed by collection of feces 12 hr after orogastric inoculation. Feces were homogenized in $ddH_2O$, diluted, and plated on MacConkey agar plates (Kanamycin 100 µg/ml).

For CFU enumeration, mice were sacrificed 10 days after *Y. pseudotuberculosis* orogastric infection. At each time point, cardiac puncture was performed and intestines, liver, and spleen were harvested, weighed, and homogenized using a potter homogenizer in $ddH_2O$ with 0.05% Triton X-100. The homogenates were diluted and plated on MacConkey agar plates (Kanamycin 100 µg/ml). All organ samples were normalized based on weight.

## Immunoblotting and co-Immunoprecipitation

The following antibodies were used for Western blots: anti-murine Mpeg1 (ab25146), anti-human Mpeg1 (ab176974), anti-Cullin-1 (Ab75817), anti-GFP (ab290), and anti-β-TRCP (Ab137674) (Abcam, Cambridge, MA); anti-GFP (sc9996) (Santa Cruz Biotechnology, Dallas, TX); anti-β-actin (2F1-1); anti-Ubiquitin (P4D1) (BioLegend, San Diego, CA, United States); anti-Ubc12 (D13D7); anti-K63-linkage specific polyubiquitin (D7A11); anti-K48-linkage specific polyubiquitin (D9D5) (Cell Signaling Technology, Danvers, MA, United States) and anti-FLAG (F-7425) (Sigma-Aldrich). Densitometry analysis was performed where indicated utilizing ImageJ software. Co-IP and ubiquitylation assays were modified from (*Shembade et al., 2010*) utilizing Dynabeads Co-immunoprecipitation kit (Thermo Fisher Scientific, Waltham, MA, United States). Whole-cell lysates were subjected to SDS-PAGE, transferred to nitrocellulose membranes, blocked in 5% milk, incubated with specific primary and secondary antibodies, then detected with SuperSignal West Pico Chemiluminescent Substrate (Thermo Fisher Scientific) by either X-ray film or Odyssey FC Imaging System (LI-COR, Lincoln, NE, United States).

## Intracellular bacterial load

The intracellular gentamicin protection assays were conducted as previously described for *S. typhimurium* experiments (*Lutwyche et al., 1998*; *Laroux et al., 2005*; *Law et al., 2010*). Briefly, 100 ng/ml of species-specific human or murine IFN-γ was added 14 hr before infection where necessary to induce uniform Perforin-2 expression. *S. typhimurium* was added as indicated. After 30 min to allow for uptake/invasion, the culture media was removed and replaced with fresh medium supplemented with gentamicin. For gentamicin protection assays, the multiplicity of infection was between 20–50 bacteria per cell to allow for sufficient uptake of bacteria.

## Extracellular bacterial load

For extracellular bacterial killing assays, mammalian cells were seeded so that >90% confluence was achieved at the time of infection. The expression of Perforin-2 was induced with species specific IFN-γ at 100 ng/ml for 14 hr prior to infection. *Y. pseudotuberculosis* or EPEC were added at an MOI of 5–10. Bacteria were allowed to attach to mammalian cell membranes for 40 min. After the attachment phase nonadherent extracellular bacteria were removed by aspiration of culture medium followed by three to five washes with PBS. Fresh culture medium was added after the final wash and cultures were incubated in 5% $CO_2$ at 37C. At selected time points the culture medium was aspirated and discarded. Adherent bacteria were recovered in $ddH_2O$ or PBS containing 0.1% (vol/vol) Triton X-100. Bacteria were serially diluted in PBS and enumerated on MacConkey or LB agar plates. Although both extracellular and intracellular bacteria are recovered by this method, we determined that the load of intracellular bacteria is numerically insignificant compared to the far higher number of adherent extracellular bacteria.

## Confocal imaging

Perforin-2 −/− MEFs or CMT93 cells were nucleofected with either murine Perforin-2 RFP or Perforin-2-KQ-RFP, plated onto coverslips and stimulated overnight with IFN-γ. Cells were washed once, and stimulated with LPS or infected with wild-type or Cif− *Y. pseudotuberculosis* as described above. Infection or LPS stimulation was allowed to proceed for 15 min upon which cells were fixed with 3% paraformaldehyde for 15 min at room temperature and counter stained with DAPI. Images were taken

on a Leica SP5 inverted confocal microscope with a motorized stage and 63× objective. Images were analyzed using Leica application suite advanced fluorescence software and ImageJ. Videos were constructed in ImageJ with 3D Projection of the confocal stacks with Y-axis rotation in the video.

## RNA interference

For murine cells, RNA interference and transfection were conducted as previously described (*Law et al., 2010*). For human cells, the aforementioned murine system was modified through utilizing three human Perforin-2-specific silencer select siRNAs purchased from Ambion (Waltham, MA, United States) Silencer Select #s61053, s47810, s61054. For Ubc12 three murine Ubc12-specific silencer select siRNAs were purchased from Ambion; Silencer Select #s75658, s75659, s75660. For murine Cullin-1 knockdown, three MISSION predesigned siRNAs were selected SASI_Mm01_00112833, SASI_Mm01_00112832, and SASI_Mm02_00322009 was utilized. Silencer select negative control #1 and 2 from Ambion were utilized as negative controls.

## Statistical analysis

Student's *t*-test, multiple t-test with Holm-Sidak multiple comparisons correction, one-way ANOVA with Bonferroni multiple comparisons test, or Kruskal–Wallis non-parametric test with Dunn's multiple comparison test was used for comparisons (GraphPad Prism Version 6.0b and SPSS 21.0 were utilized for statistical analysis).

## Acknowledgements

We acknowledge the help of Dr Y Wang in generating *Mpeg1* −/− mice and the advice and assistance of Drs V Deyev, B Watson, G Plano, N Shembade, D Fiorentino and E Fisher, University of Miami, FL. We would like to thank Drs J Bethea and L Plano, University of Miami, FL; Drs E Oswald and F Taieb, University of Toulouse, France; and Dr J Galan, Yale University, CT for providing mammalian cell lines and bacterial strains.

## Additional information

### Competing interests

RMMC: May gain royalties from commercialization of USPTO patent applications PCT/US2014/059675 and PCT/US2013/032503. KL: May gain royalties from commercialization of USPTO patent applications PCT/US2014/059675 and PCT/US2013/032503. ERP: May gain royalties from commercialization of USPTO patent applications PCT/US2014/059675 and PCT/US2013/032503. The other authors declare that no competing interests exist.

### Funding

| Funder | Grant reference | Author |
| --- | --- | --- |
| National Institutes of Health (NIH) | AI057648 | George P Munson |
| National Institutes of Health (NIH) | AI110810 | Eckhard R Podack |
| National Institutes of Health (NIH) | CA039201 | Eckhard R Podack |
| National Institutes of Health (NIH) | CA109094 | Eckhard R Podack |
| National Institutes of Health (NIH) | AI0073234 | Eckhard R Podack |
| National Institutes of Health (NIH) | AI096396 | Eckhard R Podack |
| National Institutes of Health (NIH) | AI106290 | Ryan M McCormack |
| Lois Pope Life Foundation | Developmental Fellowship | Ryan M McCormack |

The funders had no role in study design, data collection and interpretation, or the decision to submit the work for publication.

### Author contributions

RMMC, KL, ERP, GPM, Conception and design, Acquisition of data, Analysis and interpretation of data, Drafting or revising the article; MLO, Acquisition of data, Analysis and interpretation of data, Drafting or revising the article

## Author ORCIDs

George P Munson, http://orcid.org/0000-0002-3692-8199

## Ethics

Animal experimentation: This study was performed in strict accordance with the recommendations in the Guide for the Care and Use of Laboratory Animals of the National Research Council as stipulated by the National Institutes of Health. All of the animals were handled according to approved institutional animal care and use committee (IACUC) protocols (#13-233 and #12-257) of the University of Miami Miller School of Medicine.

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
