## [Decision Letter]

Thank you for sending your work entitled “Enteric pathogens deploy cell cycle inhibiting factors to block the bactericidal activity of Perforin-2” for consideration at *eLife*. Your article has been favorably evaluated by Richard Losick (Senior Editor), a Reviewing Editor, and two reviewers.

The Reviewing Editor and the reviewers discussed their comments before we reached this decision, and the Reviewing Editor has assembled the following comments to help you prepare a revised submission.

We agree that this is an interesting story that is well told, but could benefit from additional experiments to better define the mechanism and from more modest claims.

1) The authors need to tone down some of their claims, such as the following:

a) Abstract – starting statement: “Perforin-2 (MPEG1) is a principal mediator of vertebrate innate immunity that efficiently eliminates pathogenic bacteria from infected cells”;

b) Abstract – second statement: “Given the potent bactericidal activity of Perforin-2, bacterial pathogens must block its activation or otherwise circumvent its bactericidal activity in order to survive and cause disease”. It is not proven that P-2 on its own is bactericidal;

c) Abstract – concluding statement: “These studies illustrate that Perforin-2 is central in the evolutionary arms race between host and pathogen”;

d) Impact statement – “The survival and proliferation of enteric bacterial pathogens is dependent upon inactivation of Perforin-2”. This is an overstatement since they have only shown this for one pathogen;

e) The relative importance of Prf-2 compared to other mechanisms of innate immunity and bacterial defense is unclear. Moreover Prf-2 may work in conjunction with known defenses (i.e. the statement in the third paragraph of the Introduction that P-2 is not ancillary to other mechanisms is unproven).

2) Ubiquitin is 8.5 kDa. Every western blot that shows ubiquitination of P2 does not show any band shift compared to the inactive/un-ubiquitylated form. What is the explanation?

3) Cul1 and UBC12 (E3 and E2 ligases) would polyubiquitylate the cell cycle inhibitors to degrade them. Do they also polyubiquitylate P-2? If so why is it not degraded? Do proteasome inhibitors have an effect on bacterial killing? If it is monoubiquitylation on P-2 what makes the Ubi-ligase switch from poly-Ubi on cell cycle inhibitors to mono-Ubi on P-2? Is it neddylation or ubquitylation that is important? E3s give substrate specificity to Ubi-ligases. It is hard to understand the same E3 ligase Cul1 recognizes such diverse substrates. In the accompanying manuscript the authors hint that proteolytic cleavage of Prf-2 to release the active domain from the membrane may be the mechanism of Prf-2 activation – can a change in membrane localization/MW of Prf-2 be shown to be dependent on ubiquitylation?

4) It has not been demonstrated what ubiquitylation of Perforin-2 does exactly, besides correlating with bactericidal function: does it affect relocalization within the cell, proteolytic processing, etc.? These studies are simple to perform (confocal, as performed in the accompanying paper, or sequencing of native, activated Perforin-2 to determine whether it has been proteolytically processed) and would make the paper stronger.

5) Is perforin-2 being routed to proteasomal degradation? Or is this K63 signaling?

6) The EPEC strain E2438/69 in the subsection “Cell culture and bacterial organisms” and elsewhere in the manuscript is presumably E2348/69 (reverse 3 and 4), which is the prototypic EPEC strain used in the last 30 years.

In addition, in Figure 3 legend, the strain designation of E2438/69 (*cif::IS*) is not defined anywhere. Presumably this is the wild type E2348/69 (not E2438) which has a naturally occurring mutation in the *cif* gene. Or could it be that a *cif* mutation was deliberately made by insertion of an IS element?

7) CRL activity is cell cycle-regulated by neddylation. Does that mean P-2 activity is also cell cycle regulated? P-2 needs to be ubiquitylated by CRL to get active? Does that have any effect on the host cell cycle?

8) The KO mice need to be backcrossed into the background strain for multiple generations in order to be certain that differences in background aren't responsible for the effects attributed to P-2. This is a standard requirement for KO experiments.

9) It is unclear why, in the in vivo infectious studies, there is a discrepancy between fecal shedding CFUs (Figure 8–figure supplement 1) and the CFU counts associated with the intestine (Figure 10). In the intestinal counts (Figure 9), there are significant differences in the CFUs of the wild type and *cif* mutant strains in the P2 +/+ and P2 +/- mouse strains. But no differences are seen between the strains in the fecal counts. Were fecal counts tested only at 12 hours post infection while bacteria associated with the intestine counted 10 days post infection? If so, then the data in Figure 8–figure supplement 1 showing no difference in fecal shedding may be misleading. In any event, this difference should be discussed.

[Editors' note: further revisions were requested prior to acceptance, as described below.]

Thank you for resubmitting your work entitled “Enteric pathogens deploy cell cycle inhibiting factors to block the bactericidal activity of Perforin-2” for further consideration at *eLife*. Your revised article has been favorably evaluated by Richard Losick (Senior Editor) and a Reviewing Editor. The manuscript has been strikingly improved but there are some small remaining issues that need to be addressed before acceptance, as outlined below:

1) Please explain for the non-specialized reader what is NEDD8.

2) Please try to present another model: the one in Figure 14 is not clear enough.

3) Figure 5 and corresponding legend should be improved. The delta mutant should clearly be named.

---

## [Author Response]

1) The authors need to tone down some of their claims, such as the following:

a) Abstract – starting statement: “Perforin-2 (MPEG1) is a principal mediator of vertebrate innate immunity that efficiently eliminates pathogenic bacteria from infected cells”;

b) Abstract – second statement: “Given the potent bactericidal activity of Perforin-2, bacterial pathogens must block its activation or otherwise circumvent its bactericidal activity in order to survive and cause disease”. It is not proven that P-2 on its own is bactericidal;

c) Abstract – concluding statement: “These studies illustrate that Perforin-2 is central in the evolutionary arms race between host and pathogen”;

d) Impact statement – “The survival and proliferation of enteric bacterial pathogens is dependent upon inactivation of Perforin-2”. This is an overstatement since they have only shown this for one pathogen;

e) The relative importance of Prf-2 compared to other mechanisms of innate immunity and bacterial defense is unclear. Moreover Prf-2 may work in conjunction with known defenses (i.e. the statement in the third paragraph of the Introduction that P-2 is not ancillary to other mechanisms is unproven).

We have revised our manuscript and impact statement rhetoric accordingly.

2) Ubiquitin is 8.5 kDa. Every western blot that shows ubiquitination of P2 does not show any band shift compared to the inactive/un-ubiquitylated form. What is the explanation?

We now show that Perforin-2 is monoubiquitylated. Since Perforin-2-GFP is 105 kDa, monoubiquitylation increases its mass by a mere 7%. On some gels this small increase is not apparent. However, a band shift is readily apparent in newly added figures showing gels that have been run considerably longer than before. We have also noted the apparent discrepancy in our revised text (please see the subsection “Monoubiquitylation of Perforin-2 signals its cellular redistribution”).

3) Cul1 and UBC12 (E3 and E2 ligases) would polyubiquitylate the cell cycle inhibitors to degrade them. Do they also polyubiquitylate P-2? If so why is it not degraded? Do proteasome inhibitors have an effect on bacterial killing?

In our revised manuscript we show that Perforin-2 is monoubiquitylated (Figure 11). We also show that ubiquitylation is required for the trafficking of Perforin-2 (Figures 12 and 13). Based upon this new data and analyses, it is unlikely that the proteasome is involved in regulating Perforin-2 activity.

If it is monoubiquitylation on P-2 what makes the Ubi-ligase switch from poly-Ubi on cell cycle inhibitors to mono-Ubi on P-2?

This is an interesting mechanistic question. However, the mono/poly determinant does not appear to be well understood within the CRL field and is beyond the scope of our current study.

Is it neddylation or ubquitylation that is important?

They are both equally important. Perforin-2 activity/trafficking is dependent upon ubiquitylation by a CRL. The activity of the CRL is dependent upon neddylation. Greater clarity is provided by a new summary figure (Figure 14) that outlines the key mechanistic steps in the Perforin-2 dependent killing pathway.

E3s give substrate specificity to Ubi-ligases. It is hard to understand the same E3 ligase Cul1 recognizes such diverse substrates.

Technically it is substrate receptors, such as bTrCP, that provide substrate specificity. A conserved domain within substrate receptors, the F-box, allows them to interact – often through adaptors – with cullins. In this way a large repertoire of substrate receptors may interact with a small number of cullins. Previously we did not sufficiently explain these somewhat complicated systems. This deficiency is addressed in Figure 14 and elsewhere within the text as appropriate.

In the accompanying manuscript, the authors hint that proteolytic cleavage of Prf-2 to release the active domain from the membrane may be the mechanism of Prf-2 activation – can a change in membrane localization/MW of Prf-2 be shown to be dependent on ubiquitylation?

In our accompanying manuscript we show that Perforin-2 is cleaved and map the approximate cleavage sites. Moreover, the cleavage products are associated with bacteria. These new findings are discussed within our revised manuscript (Discussion). In future studies we will determine if cleavage occurs before or after ubiquitylation. However, we do demonstrate in Figures 12 and 13 and Videos 1 and 2 that Perforin-2 relocalization is dependent on ubiquitylation.

4) It has not been demonstrated what ubiquitylation of Perforin-2 does exactly, besides correlating with bactericidal function: does it affect relocalization within the cell, proteolytic processing, etc.? These studies are simple to perform (confocal, as performed in the accompanying paper, or sequencing of native, activated Perforin-2 to determine whether it has been proteolytically processed) and would make the paper stronger.

The suggested studies have been added to our revised manuscript (Figures 12 and 13). We can now state definitively that ubiquitylation is necessary for the relocalization of Perforin-2.

5) Is perforin-2 being routed to proteasomal degradation? Or is this K63 signaling?

Neither. Newly added data shows that Perforin-2 is monoubiquitylated and that monoubiquitylation serves as a routing signal (Figures 11, 12 and 13).

6) The EPEC strain E2438/69 in the subsection “Cell culture and bacterial organisms” and elsewhere in the manuscript is presumably E2348/69 (reverse 3 and 4), which is the prototypic EPEC strain used in the last 30 years.

*In addition, in*
Figure 3
*legend, the strain designation of E2438/69 (*cif::IS*) is not defined anywhere. Presumably this is the wild type E2348/69 (not E2438) which has a naturally occurring mutation in the* cif *gene. Or could it be that a* cif *mutation was deliberately made by insertion of an IS element?*

This was a typo. As indicated by the reviewers the correct strain name is E2348/69. As per Marches et al., this strain harbors a natural IS within *cif*. We have revised our figures (Figures 5 and 6) and text for clarity.

7) CRL activity is cell cycle-regulated by neddylation. Does that mean P-2 activity is also cell cycle regulated? P-2 needs to be ubiquitylated by CRL to get active? Does that have any effect on the host cell cycle?

We do not believe that the first sentence is entirely correct. Rather, it is our understanding from the literature that a CRL regulates cell cycling and not the other way around. The ability of the CRL to recognize its substrate, via a F-box substrate receptor, is often dependent upon substrate modification, such as phosphorylation. In either case, we have no evidence that Perforin-2 is cell cycle regulated.

8) The KO mice need to be backcrossed into the background strain for multiple generations in order to be certain that differences in background aren't responsible for the effects attributed to P-2. This is a standard requirement for KO experiments.

In our initial study the Perforin-2 mutation was backcrossed 7-10 times, a point that is now made clear (subsecton “Generation of Mpeg1 knockout mice”). However, we agree that is not sufficient to remove doubts as per the cause of the phenotype. Therefore we now include additional animal studies with genetically pure 129X1/SvJ mice (Figure 9). The results with these new mice are similar to those previously reported (Figure 9). Therefore we are confident that the reported phenotypes are due to Perforin-2 deficiency and not the result of passenger mutations or differences between strains.

*9) It is unclear why, in the in vivo infectious studies, there is a discrepancy between fecal shedding CFUs (Figure 8–figure supplement 1) and the CFU counts associated with the intestine (*Figure 10*). In the intestinal counts (*Figure 9*), there are significant differences in the CFUs of the wild type and* cif *mutant strains in the P2 +/+ and P2 +/- mouse strains. But no differences are seen between the strains in the fecal counts. Were fecal counts tested only at 12 hours post infection while bacteria associated with the intestine counted 10 days post infection? If so, then the data in Figure 8–figure supplement 1 showing no difference in fecal shedding may be misleading. In any event, this difference should be discussed.*

The discrepancy is indeed due to timing – fecal pellets were collected 12 hours post inoculation while organ loads were done at day 10. We agree that the fecal shedding at 12 hours is confusing and in retrospect, the data is unnecessary. As such, we have removed it from our revised manuscript.

[Editors' note: further revisions were requested prior to acceptance, as described below.]

1) Please explain for the non-specialized reader what is NEDD8.

We have added “the ubiquitin-like protein NEDD8” to our Abstract. In the Introduction, we also added the following: “NEDD8, an 8.6 kDa member of the ubiquitin family of proteins (UniProt entry Q15843, Pfam identifier PF00240)…”.

*2) Please try to present another model: the one in*
Figure 14
*is not clear enough.*

We have substantially modified Figure 14 so that it focuses on the CRL complex and the relationship between neddylation and ubiquitylation of Perforin-2. These are now shown as sequential steps and the mechanism of inhibition by Cif is clearly shown. We have omitted a depiction of Perforin-2 trafficking as this is easily explained in the text.

We hope that this more refined figure is sufficiently clear. If not, please elaborate on the problematic aspects of the figure.

*3)*
Figure 5
*and corresponding legend should be improved. The delta mutant should clearly be named.*

Figure 5 has been revised and now uses standard genetic nomenclature. The legend has been modified accordingly. We have also slightly modified Figure 6 and Figure 7 so that Figures 5, 6 and 7 are consistent with each other. Specifically we changed *cifYp::aadA* so that if reads *cif::aadA*.